# SOS1 tonoplast neo-localization and the RGG protein SALTY are important in the extreme salinity tolerance of *Salicornia bigelovii*

Octavio R. Salazar [1,2,3], Ke Chen [4], Vanessa J. Melino [1,2], Muppala P. Reddy[1,2], Eva Hřibová [5], Jana Čížková [5], Denisa Beránková [5], Juan Pablo Arciniegas Vega [1,2], Lina María Cáceres Leal [1,2], Manuel Aranda [1,3], Lukasz Jaremko [1], Mariusz Jaremko [1], Nina V. Fedoroff [6], Mark Tester [1,2] ✉ & Sandra M. Schmöckel [7]

The identification of genes involved in salinity tolerance has primarily focused on model plants and crops. However, plants naturally adapted to highly saline environments offer valuable insights into tolerance to extreme salinity. *Salicornia* plants grow in coastal salt marshes, stimulated by NaCl. To understand this tolerance, we generated genome sequences of two *Salicornia* species and analyzed the transcriptomic and proteomic responses of *Salicornia bigelovii* to NaCl. Subcellular membrane proteomes reveal that SbiSOS1, a homolog of the well-known SALT-OVERLY-SENSITIVE 1 (SOS1) protein, appears to localize to the tonoplast, consistent with subcellular localization assays in tobacco. This neo-localized protein can pump $Na^+$ into the vacuole, preventing toxicity in the cytosol. We further identify 11 proteins of interest, of which SbiSALTY, substantially improves yeast growth on saline media. Structural characterization using NMR identified it as an intrinsically disordered protein, localizing to the endoplasmic reticulum *in planta*, where it can interact with ribosomes and RNA, stabilizing or protecting them during salt stress.

Salinity is one of the major factors limiting crop productivity, with most crops being relatively salt sensitive and significantly affected when exposed to NaCl in the range of 50–200 mM[1]. Currently, at least 20% of irrigated lands are affected by salinity[2], resulting in reduced crop production. Furthermore, soil salinization is increasing yearly[3,4]. With the world's population projected to grow from 7.7 to 9.7 billion people by 2050[5], the demand for increased agricultural production will likely require expanding crop production on marginal land, including saline soils. Studies in salinity tolerance have primarily focused on

plants that grow at low concentrations of salt[6], which includes all major crops. However, plants that thrive in highly saline environments have the potential to be valuable in the development of salt tolerant crops, as they already possess adaptations that allow them to thrive in saline environments.

The genus *Salicornia* belongs to the subfamily Salicornioideae in the Amaranthaceae family and plants of this genus grow in coastal salt marshes and inland salt lakes throughout the world[7]. They are amongst the most salt tolerant of all plant species and can thrive in

[1]Biological and Environmental Sciences & Engineering Division (BESE), King Abdullah University of Science and Technology (KAUST), Thuwal 23955-6900, Kingdom of Saudi Arabia. [2]Center for Desert Agriculture, King Abdullah University of Science and Technology (KAUST), Thuwal 23955-6900, Kingdom of Saudi Arabia. [3]Red Sea Research Center, King Abdullah University of Science and Technology (KAUST), Thuwal 23955-6900, Kingdom of Saudi Arabia. [4]Rice Research Institute, Guangdong Academy of Agricultural Sciences, Guangzhou 510640, China. [5]Institute of Experimental Botany of the Czech Academy of Sciences, Centre of Plant Structural and Functional Genomics, Šlechtitelů 31, 77900 Olomouc, Czech Republic. [6]Department of Biology, Penn State University, University Park, PA 16801, US. [7]Department Physiology of Yield Stability, Institute of Crop Science, University of Hohenheim, Fruwirthstr. 21, 70599 Stuttgart, Germany. ✉e-mail: mark.tester@kaust.edu.sa

environments with salinities even higher than seawater[8,9]. Notably, *Salicornia* plants can accumulate high concentrations of $Na^+$ in their photosynthetically active succulent shoots while avoiding ion toxicity[8,10,11], suggesting highly efficient ion compartmentalization processes in cells through the action of specialized transporters[12–15]. Moreover, *Salicornia* growth is promoted by the addition of NaCl[11,16–18], making it a particularly interesting genus for the study of salinity tolerance.

In this study, we generated draft genome sequences for two Salicornioideae species: *Salicornia bigelovii*, endemic to Mexico and the United States; and *Salicornia europaea*, endemic to Europe, Northern Africa, the Middle East, and Central Asia. We then focused on *S. bigelovii* and investigated its phenotypic, transcriptomic, and proteomic responses to different concentrations of NaCl, generating organellar membrane proteomes to enable allocation of membrane proteins to particular organelles. The protein SALT-OVERLY-SENSITIVE 1 (SOS1) unexpectedly appears to be localized to the vacuolar membrane, suggesting its neo-localization in *S. bigelovii*. Based on the transcriptomic and proteomic responses, we selected 45 candidate genes likely to contribute to salinity tolerance in *S. bigelovii* and found 11 altering salinity tolerance in a yeast expression system. We also further investigated a protein we named SALTY, which greatly improved salinity tolerance when expressed in yeast. Using nuclear magnetic resonance (NMR), SALTY was found to be an intrinsically disordered protein. This protein was localized to the endoplasmic reticulum (ER) where it is likely to interact with ribosomes and RNA.

## Results

### NaCl stimulates growth of *S. bigelovii*

We analyzed the phenotypic responses of *S. bigelovii* to NaCl by treating 5-week-old plants with 0, 50, 200, and 600 mM NaCl. After 1 week, plants treated with NaCl were taller than plants treated with 0 mM NaCl, and plants treated with 200 and 600 mM NaCl developed lateral branches (Fig. 1a, b). The beneficial effects of NaCl became increasingly apparent with time, and 6 weeks after treatment, all plants supplemented with NaCl were significantly taller and displayed a lighter green hue (Fig. 1c, d). Plants treated with 600 mM NaCl were shorter than those treated with 50 or 200 mM NaCl, but had thicker shoots (Fig. 1e, f), suggesting that a concentration of NaCl somewhere between 200 mM–600 mM NaCl is optimal for the growth of *S. bigelovii*, this is in agreement with previous studies[11,16–19].

### *S. bigelovii* accumulates large amounts of $Na^+$ and $Cl^-$ in its shoots

High concentrations of NaCl are usually detrimental to plants, as ion accumulation in shoots may lead to cell toxicity and premature senescence[1,20,21]. In plant cells, $K^+$ is usually at least 10 times more abundant than $Na^+$, with cells experiencing $Na^+$ toxicity when the $Na^+/K^+$ is altered in favor of $Na^+$[22], as $Na^+$ and $K^+$ are thought to compete for functionally important binding sites inside the cell[23–25].

$Na^+$ and $Cl^-$ concentrations increased in shoots and roots with increasing NaCl treatment (Fig. 1g, h). We found $Na^+$ and $Cl^-$ concentrations in excess of 750 mM in the shoots and 400 mM in the roots of *S. bigelovii* treated with 600 mM NaCl, which accounted for 60% of shoot and 10% of root dry mass (Supplementary Fig. 1). $K^+$ concentrations decreased with increasing NaCl treatment (Fig. 1i) and $Na^+/K^+$ was greater than 1 in shoots in all treatments, being over 20 times greater in plants treated with 600 mM NaCl (Supplementary Fig. 1). In roots, $Na^+$ concentrations were less than two-fold higher than $K^+$ at 600 mM NaCl. Shoot sap osmolality increased with NaCl treatment (Supplementary Fig. 1). Water content was higher in shoots compared with roots and increased with NaCl treatment (Fig. 1j). It was positively correlated with both $Na^+$ and $Cl^-$ and negatively correlated with $K^+$ (Supplementary Fig. 2), suggesting that $Na^+$ and $Cl^-$ could be used as osmolytes to facilitate water retention in the shoots[26].

### Genome sequencing and assembly

To gain insight into the molecular mechanisms driving the outstanding salt tolerance of *Salicornia*, we first generated genome assemblies for two *Salicornia* species: *S. bigelovii*, an autotetraploid ($2n = 4x = 36$)[7], and *S. europaea*, a diploid ($2n = 2x = 18$)[7,27]. Here we used PacBio HiFi sequencing to generate near chromosome-scale genome assemblies, resulting in haploid genome assemblies of 2,026 and 517 Mb for *S. bigelovii* and *S. europaea*, respectively (Supplementary Table 1). These assemblies encompassed > 99% of the genome in 33 contigs for *S. bigelovii* and 20 for *S. europaea*. The genome assemblies were in line with flow cytometry-based genome size estimates and showed high assembly completeness[28,29] (Supplementary Figs. 3 and 4). *S. europaea* flow cytometry values were similar to what has been reported for diploid *Salicornia* in the Gulf of Trieste[30], yet interestingly, *S. bigelovii* shows a larger genome of almost double the size of previously reported tetraploid *Salicornia*. One of the potential causes for this could be the increased percentage of repetitive elements in the genome of *S. bigelovii* (Supplementary Table 1), but further investigation is required. The genomes were predicted to contain 63,843 and 28,794 genes for *S. bigelovii* and *S. europaea*, respectively. The predicted proteins were annotated against Swiss-Prot, TrEMBL[31], NCBI NR[32], MapMan[33,34], and KEGG[35] databases (Supplementary Data 1 and 2). The completeness of the gene prediction was assessed with BUSCO[36], resulting in completeness scores of 96.3% and 95.3% for *S. bigelovii* and *S. europaea*, respectively (Supplementary Table 1).

### Gene orthology analysis

To explore possible genetic adaptations of *Salicornia* to salinity we identified groups of orthologous proteins across representative genomes of five plant families (Brassicaceae, Amaranthaceae, Fabaceae, Solanaceae, and Poaceae) with OrthoMCL[37], aiming to identify groups of proteins specific to different plant clades. A total of 38,356 orthogroups were identified, composed of 355,771 proteins (Supplementary Data 3). Phylogenetic reconstruction based on 1377 orthologs, resulted in a resolved phylogeny by family with the genus *Salicornia* forming a subclade within the family Amaranthaceae (Supplementary Fig. 5). A total of 9,470 orthogroups were shared across all families (Fig. 2a), and gene ontology (GO)[38,39] enrichment analyses of unique orthogroups revealed enrichments specific to each family (Supplementary Data 3). Orthogroups unique and present in all Amaranthaceae, were enriched in GO terms such as 'response to reactive oxygen species', 'positive regulation of sodium ion transport', 'clustering of voltage-gated sodium channels', and 'positive regulation of protein targeting to membrane' (Supplementary Data 3), which could in part explain the high salt tolerance of plants in this family.

When looking at orthogroups within the Amaranthaceae, the majority of the orthogroups were found in all species, sharing 11,350 orthogroups (Fig. 2b). GO enrichment analysis of unique orthogroups in *Salicornia* species identified enrichment of terms related to light, in the form of 'response to UV-A', 'response to far red light', and 'response to high light intensity'; to 'ion binding', in the form of zinc and manganese; to toxin production, in the form of 'toxin catabolic and metabolic processes'; and to 'mechanically gated ion channel', particularly in the form of calcium channels. These calcium channels could be involved in calcium wave signaling during NaCl stress[40–42]. Nevertheless, no orthogroup was enriched for $Na^+$ transport.

We then looked at possible expansions of sodium/proton exchanger (NHX) proteins of the cation/proton antiporter (CPA) family, which are commonly associated with salt tolerance[1,20,43]. Phylogenetic reconstruction of NHX proteins resulted in clustering by type and family (Fig. 2c and Supplementary Data 4). We identified 6 NHX proteins in *S. bigelovii* and *S. europaea*, of which 4 resembled the *Arabidopsis thaliana* vacuolar NHX1-4[44–46], one resembled the endosomal NHX6[47], and one the plasma membrane NHX7, also known as SALT-OVERLY-SENSITIVE 1 (SOS1)[48] (Fig. 2c). Both contained only one

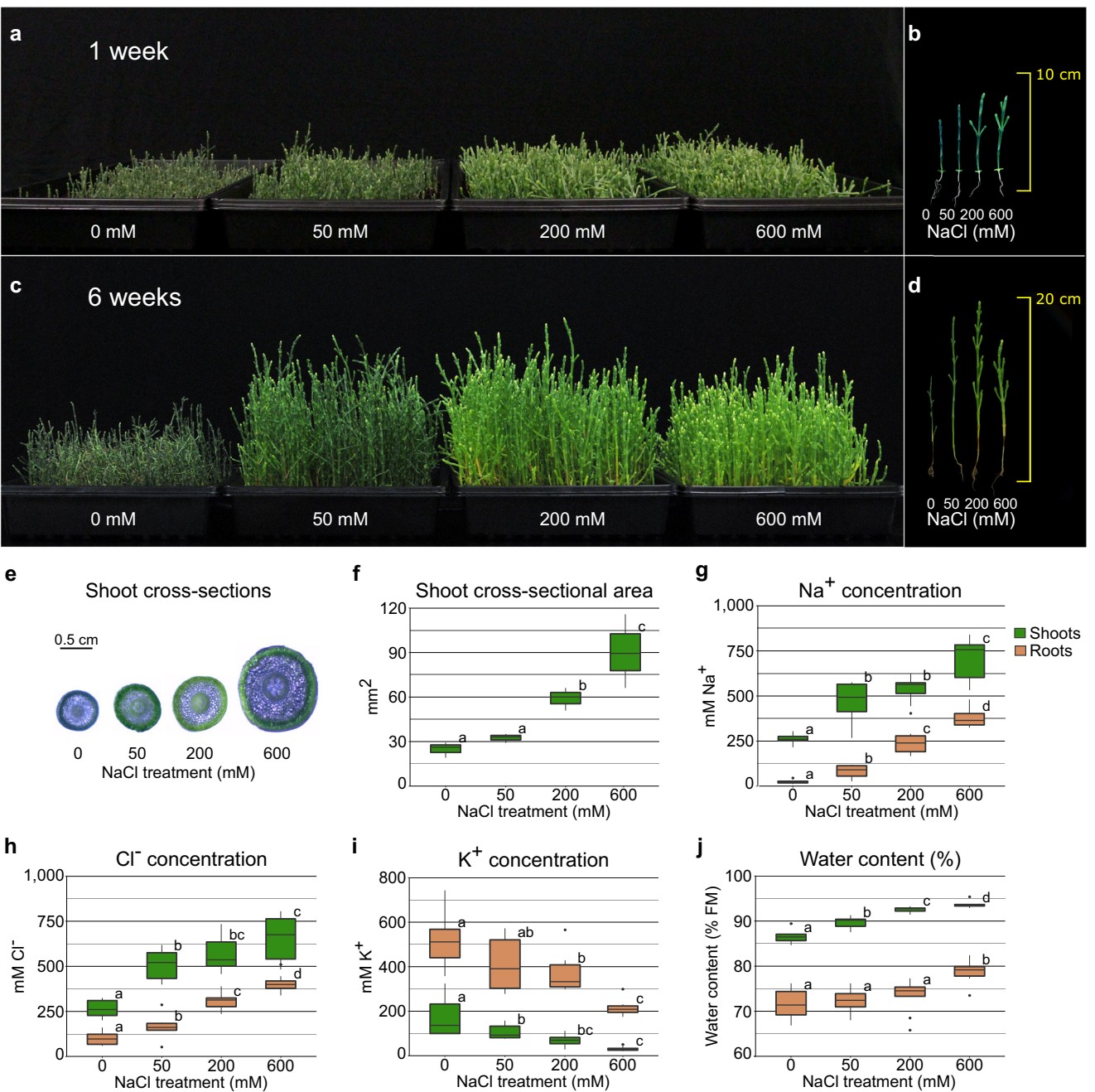

**Fig. 1 | *S. bigelovii* growth and ion accumulation under different salinities.** Plants were grown for 5 weeks prior to the addition of 0, 50, 200, or 600 mM NaCl. **a** Plants 1 week after treatment. **b** Length of individual plants 1 week after treatment. **c**, Plants 6 weeks after treatment. **d** Length of individual plants 6 weeks after treatment. **e** Shoot cross-sections. **f** Shoot cross-sectional area **g**, Na$^+$ concentration in shoots and roots. **h** Cl$^-$ concentration in shoots and roots. **i** K$^+$ concentration in shoots and roots. **j** Water content in shoots and roots. $n = 3$ (**f**) and $n = 9$ (**g**–**j**) biologically independent samples per treatment. Mean differences were compared within each tissue through two-sided $t$-tests and the FDR was controlled with the Benjamini-Hochberg procedure at an $\alpha = 0.05$, significant differences are indicated as different letters. Boxes: represent the interval between the 25th and 75th percentile with the median shown as a horizontal line. Whiskers: represent the maximum and minimum values or 1.5-fold the interquartile range when the data point is outside this range. Outliers are shown as black dots. Plant images (**a**) and (**c**) are representative images of three tray replicates.

endosomal NHX protein and no homolog for the Li$^+$/H$^+$ antiporter NHX8[49] was identified in either *Salicornia* genome, similarly to what has been observed in sugar beet[50], another Amaranthaceae. Furthermore, proteins resembling NHX8 from families other than Brassicaceae, are small in size, appear to be truncated, and are usually localized next to an ORF coding for a complementary protein, that when combined, resemble SOS1. Further inspection would be required to identify if this is the result of the degeneration of a previous duplication event, or if it is an artifact of gene prediction. These results show no

*NHX* gene expansion in *Salicornia*, indicating that its outstanding salt tolerance does not come from an increased repertoire of NHX proteins. We then looked at gene expansions in *Salicornia*, defined as having at least twice the number of proteins in *S. bigelovii* and *S. europaea* when compared to any other species within the orthogroup. GO enrichment analysis of these proteins revealed enrichments in terms related to transposable elements and iron transport, but no term was related to sodium transport (Supplementary Data 3). All together, these results suggest that *Salicornia* could have a different repertoire

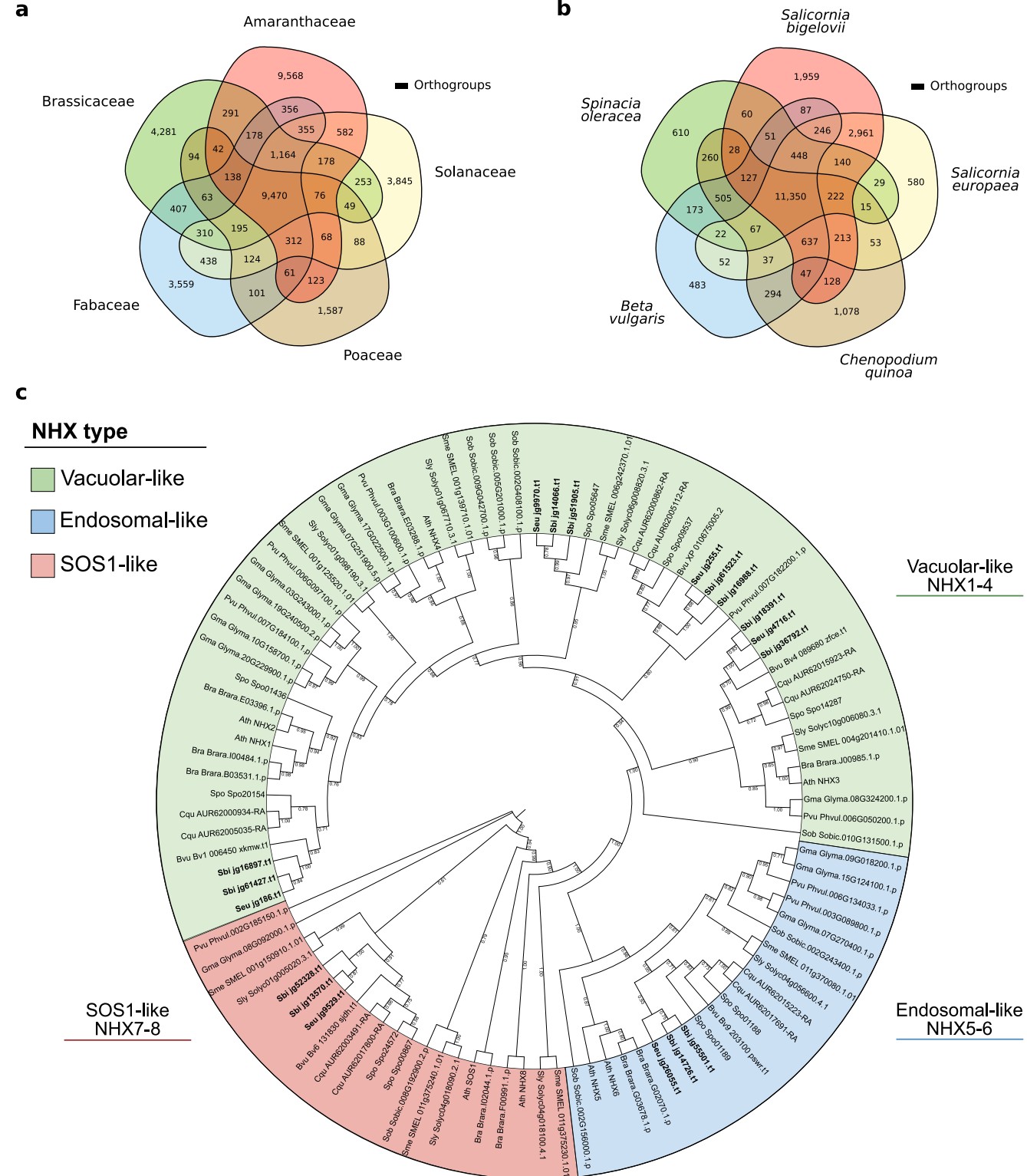

**Fig. 2 | Conservation of orthologous proteins across families and species.**
**a** Number of orthogroups shared across the families Brassicaceae (*Arabidopsis thaliana*, *Brassica rapa*), Amaranthaceae (*Beta vulgaris*, *Chenopodium quinoa*, *S. bigelovii*, *S. europaea*, *Spinacia oleracea*), Fabaceae (*Glycine max*, *Phaseolus vulgaris*), Solanaceae (*Solanum lycopersicum*, *Solanum melongena*), and Poaceae (*Sorghum bicolor*). **b** Number of orthogroups shared across the Amaranthaceae species: *Beta vulgaris*, *Chenopodium quinoa*, *S. bigelovii*, *S. europaea*, and *Spinacia oleracea*. **c** Phylogenetic reconstruction of NHX proteins. In bold, *Salicornia* proteins. The phylogenetic tree was generated with RAxML-NG[166] and visualized with iTOL[168], values at branching nodes represent transfer bootstrap expectation (TBE)[167] values based on 1,000 replicates.

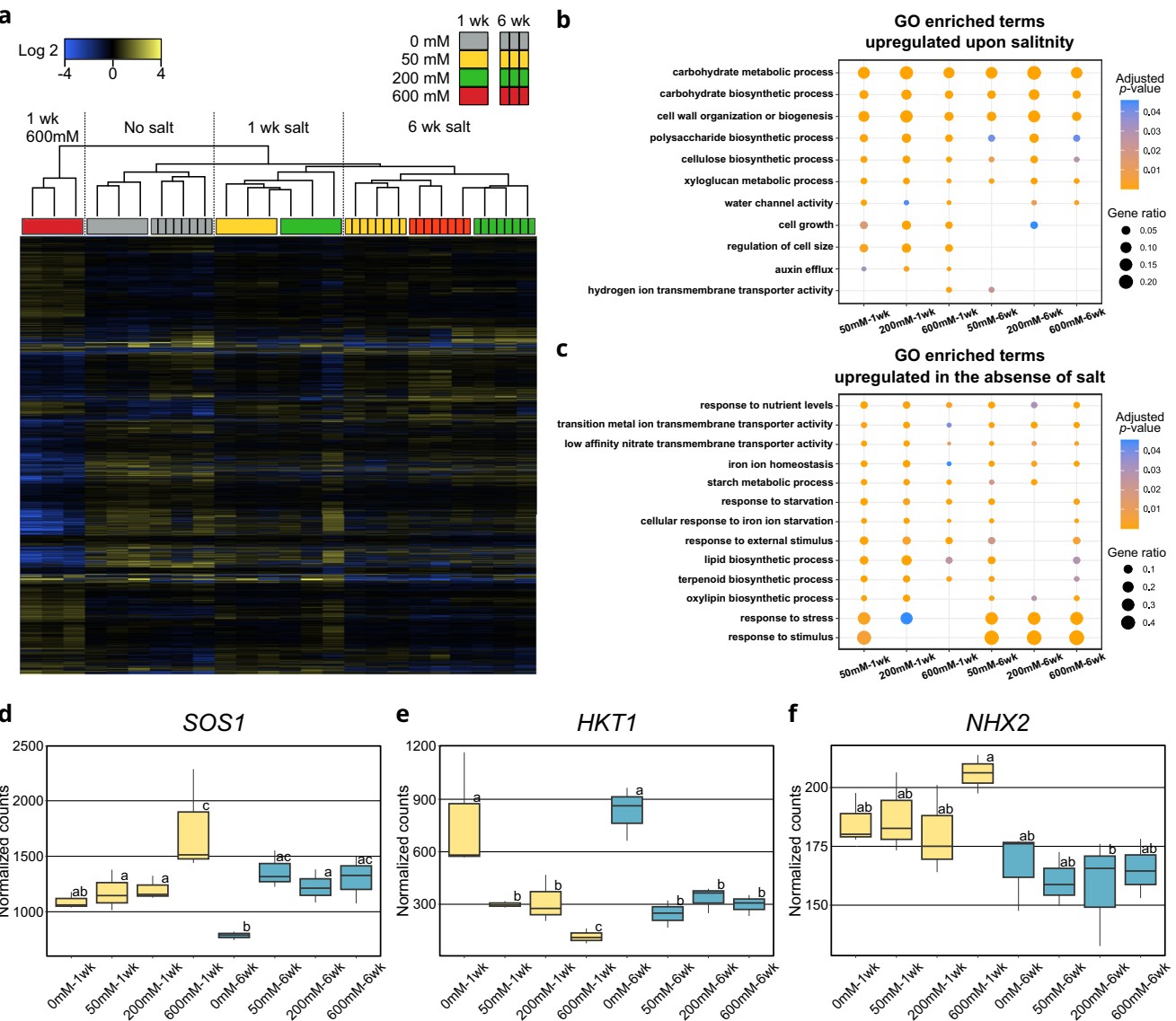

**Fig. 3 | *S. bigelovii* transcriptional responses to NaCl. a** Hierarchical clustering of differentially expressed genes in shoots of *S. bigelovii* plants treated with 0, 50, 200, and 600 mM NaCl for 1 and 6 weeks. **b**–**c** Representative GO enriched terms relative to 0 mM NaCl treated plants. **b** upregulated upon the addition of NaCl, **c**, upregulated in 0 mM NaCl treated plants. **d**–**f** Gene expression of proteins commonly associated with Na⁺ transport. Yellow, plants treated for 1 week; Blue, plants treated for 6 weeks. **d** *SOS1*, **e** *HKT1*, **f**, *NHX2*. Gene expression differences were compared within DESeq2 at an $\alpha = 0.05$, significant differences are indicated as different letters. $n = 3$ biologically independent samples per treatment. Boxes: represent the interval between the 25th and 75th percentile with the median shown as a horizontal line. Whiskers: represent the maximum and minimum values or 1.5-fold the interquartile range when the data point is outside this range. Outliers are shown as black dots.

of currently unknown genes involved in salinity tolerance or that it has fine-tuned the same genes found in other species to cope with extremely saline environments.

### NaCl induces expression of genes associated with growth

To gain insights into the responses of *S. bigelovii* to NaCl, we studied the transcriptomic responses in shoots of plants exposed to 0, 50, 200, and 600 mM NaCl for 1 and 6 weeks. We identified 11,799 differentially expressed genes between at least two treatments (Fig. 3a, Supplementary Table 2, Supplementary Data 4). Plants that were not treated with NaCl clustered together regardless of their age, while plants treated with NaCl clustered according to their age and treatment (Fig. 3a). This could be caused by the limited growth that plants displayed without the addition of NaCl and by the developmental responses that occurred upon NaCl treatment (Fig. 1). Plants treated with 600 mM NaCl for 1 week clustered separately from any time point

or treatment and had the highest number of differentially expressed genes. This suggests a shock response to the high concentration of NaCl, as 6 weeks after treatment, their gene expression became similar to those treated with 50 and 200 mM NaCl, indicating that plants had adapted to the high concentration of NaCl (Fig. 3a).

GO enrichment analysis of genes upregulated in plants treated with NaCl revealed enrichment in terms related to growth, such as 'carbohydrate biosynthetic and metabolic processes', 'cell wall biogenesis and modification', 'regulation of cell size', and 'water transport' (Fig. 3b and Supplementary Data 5). This is in agreement with the increased growth, cell size, and water content observed in *S. bigelovii* plants upon the addition of salt (Fig. 1). Conversely, plants grown in the absence of NaCl had enriched terms such as 'response to starvation', 'response to stress', 'response to nutrient levels', 'iron and nitrate transport', 'starch metabolic process', and 'lipid, terpenoid and oxylipin biosynthetic processes', indicating that plants grown without salt

**Table 1 | Total number of proteins identified and number of marker proteins per organelle for *S. bigelovii* membrane preparations from plants treated with 0, 50, 200, or 600 mM NaCl**

| Marker type/NaCl | 0 mM | 50 mM | 200 mM | 600 mM |
|---|---|---|---|---|
| Endoplasmic reticulum | 18 | 18 | 20 | 17 |
| Mitochondrion | 35 | 35 | 36 | 35 |
| Plasma membrane | 22 | 21 | 21 | 21 |
| Chloroplast | 30 | 23 | 32 | 25 |
| Tonoplast | 18 | 17 | 18 | 15 |
| **Total markers** | **123** | **114** | **127** | **113** |
| Unallocated proteins | 1535 | 1462 | 1461 | 1512 |
| Total proteins | 1658 | 1576 | 1588 | 1625 |

were under stress and suffered nutritional imbalance (Fig. 3c and Supplementary Data 5). MapMan[33,34] pathway analysis revealed similar results (Supplementary Tables 3 and 4), but also showed an enrichment in the downregulation of genes coding for receptor proteins with domain of unknown function 26 (DUF26). This downregulation was greatly overrepresented in plants treated with 600 mM NaCl for 1 week. Downregulation of the DUF26 protein, OsRMC, in rice increased survival of rice plants when exposed to severe NaCl treatments[51]. The increased downregulation of DUF26 proteins in *S. bigelovii* when exposed to high concentrations of NaCl could have a similar effect as in rice, increasing survival in high salinity.

We then looked at the expression of genes encoding transporter proteins commonly associated with salinity tolerance and found contrasting gene expression profiles (Fig. 3d–f). *SOS1* was upregulated in plants treated with NaCl, the high-affinity potassium transporter (*HKT1*)[52,53] expression was downregulated upon NaCl treatment, and *NHXs* gene expression remained constant throughout treatments (Supplementary Data 4). Previous studies in *Salicornia dolichostachya*, based on quantitative Reverse Transcription (qRT)-PCR, suggested *HKT1* to be absent from *Salicornia*[54]. However, we found *HKT1* to be present in both *Salicornia* species.

## *S. bigelovii* organellar membrane proteome

We hypothesized that specific ion transporters must play key roles in preventing ion toxicity in *S. bigelovii* cells, extruding Na$^+$ from the cytosol and compartmentalizing it in the vacuole. Proteomic analyses of specific membrane systems are possible as membranes from different organelles have different densities, allowing for their separation through ultracentrifugation[55,56] (Supplementary Fig. 6). The separation does not result in pure isolates, but in fractions enriched with membranes from specific organelles[57,58]. We isolated membrane-enriched fractions from shoots of *S. bigelovii* plants treated with 0, 50, 200, and 600 mM NaCl for 6 weeks, resulting in 12 different density fractions from a continuous sucrose gradient covering a specific gravity range, from 1.08 to 1.19 relative to water, and a maximum variation of 0.8% sucrose or 0.003 specific gravity within fractions (Supplementary Figs. 6 and 7).

Membrane proteins were digested using the gel-free filter-aided sample preparation protocol (FASP)[59], resulting in the identification of more than 1,500 protein clusters per treatment (Table 1) and a total of 2118 unique protein clusters. Markers from 5 different organelles were identified based on previous proteomic studies in Arabidopsis[57,60–64].

Organellar membrane profiles were generated by assessing the relative abundance of marker proteins at the different density fractions with pRoloc[65–70]. To validate the in silico organellar profiles generated with pRoloc, western blots were carried out across the 12 different density fractions against marker proteins for the thylakoid (PsbA-D1), plasma membrane (H$^+$-ATPase) and tonoplast (V-ATPase) (Fig. 4a, Supplementary Fig. 8). In silico organellar protein profiles

coincided with the results obtained from western blots (Fig. 4a, b), indicating a good organelle profile accuracy of our in silico profiles. To allocate a subcellular localization to the unclassified proteins, their relative abundances were compared against organellar profiles across the 12 different density fractions (Fig. 4c and Supplementary Data 6). Due to similarities in their densities, separation between the plasma membrane and endoplasmic reticulum (ER), and between the chloroplast and mitochondrion, were difficult. The tonoplast fractions were distinctly different from other fractions (Fig. 4c) and had a greater relative density compared to that found in Arabidopsis[56] and other glycophyte species, where the tonoplast is the least dense membrane. This apparently higher density of the tonoplast in *S. bigelovii* is supported not only by the western blot analysis (Fig. 4a), but also by the abundance profiles of tonoplast marker proteins such as vacuolar ATPase subunits and the vacuolar pyrophosphatase (Fig. 4c, Supplementary Figs. 9–11, and Supplementary Data 6). This higher density may be due to higher levels of membrane proteins in the tonoplast, perhaps related to the very high concentration of ions accumulating in the vacuoles of *Salicornia* shoots.

## Possible neo-localization of SOS1 to the tonoplast in *S. bigelovii*

Exploration of the organellar membrane proteome revealed the localization of SOS1, also known as NHX7, to the tonoplast (Fig. 4c–g). SOS1 was first identified in Arabidopsis[48,71] and has been described as a Na$^+$/H$^+$ antiporter localized to the plasma membrane, extruding Na$^+$ from the cytosol into the apoplast[48,72–74]. SbiSOS1 is predicted to be 1159 amino acids in length, the same as the predicted SOS1 sequence in *S. europaea* and the reported SOS1 sequence for *S. brachiata*[75]. SbiSOS1 clustered with SOS1-like proteins (Fig. 2c) and shows a 63% sequence identity to Arabidopsis SOS1 and only a 27% identity to other NHX proteins. Similar sequence identities can be observed between the SOS1 protein of Arabidopsis and those of Quinoa, Rice, and Tomato (61-63%). Protein abundance profiles show that SbiSOS1 localizes to the tonoplast. This was consistent across all treatments, and in all instances, it had a high allocation score to the tonoplast (Fig. 4d–g, Supplementary Figs. 9–11, Supplementary Data 6). Furthermore, it was closely distributed with tonoplast marker proteins such as: vacuolar ATPase subunits a, b, c, d; tonoplast intrinsic protein TIP1-3; and the vacuolar pyrophosphatase (Fig. 4c, Supplementary Figs. 9–11). SbiSOS1 was more abundant in fractions 4-6 (Fig. 4d–g), following the abundance pattern of tonoplast marker proteins and different to plasma membrane markers (Fig. 4a,b). SbiSOS1 was amongst the most abundant membrane proteins in shoots of *S. bigelovii*, together with the plasma membrane H$^+$-ATPase, the vacuolar V-ATPases, PIPs, and TIPs (Supplementary Data 6), and its abundance did not change with treatment.

To assess the possible neo-localization of SbiSOS1 to the tonoplast, we expressed SbiSOS1 and Arabidopsis SOS1 (AtSOS1) fused with the reporter protein eGFP and assessed their colocalization with the plasma membrane AtPIP1;4 and the tonoplast marker VAMP711[76] in tobacco leaves (Fig. 5). SbiSOS1 colocalizes with the tonoplast marker (Fig. 5a–c, Supplementary Figs. 12–14), supporting the neo-localization hypothesis. This tonoplast localization was observed in both N and C-terminal eGFP fusions (Supplementary Figs. 13 and 14). However, SbiSOS1 was also observed to colocalize with AtPIP1;4 to the plasma membrane in tobacco cells (Fig. 5d–f). This dual localization of SbiSOS1 to both the tonoplast and the plasma membrane in tobacco cells, may suggest for a mechanism in *Salicornia* to reduce cytosolic salt accumulation by transporting Na$^+$ into the vacuole and by transporting it outside of the cell. Nevertheless, the observed plasma membrane localization could also be the result of excess protein being targeted to the plasma membrane due to overexpression with the 35 S promoter, as the secretory pathway is the default pathway for membrane protein localization[77]. AtSOS1 colocalized with AtPIP1;4 in the plasma membrane (Fig. 5g–l and Supplementary Fig. 12). The expression and

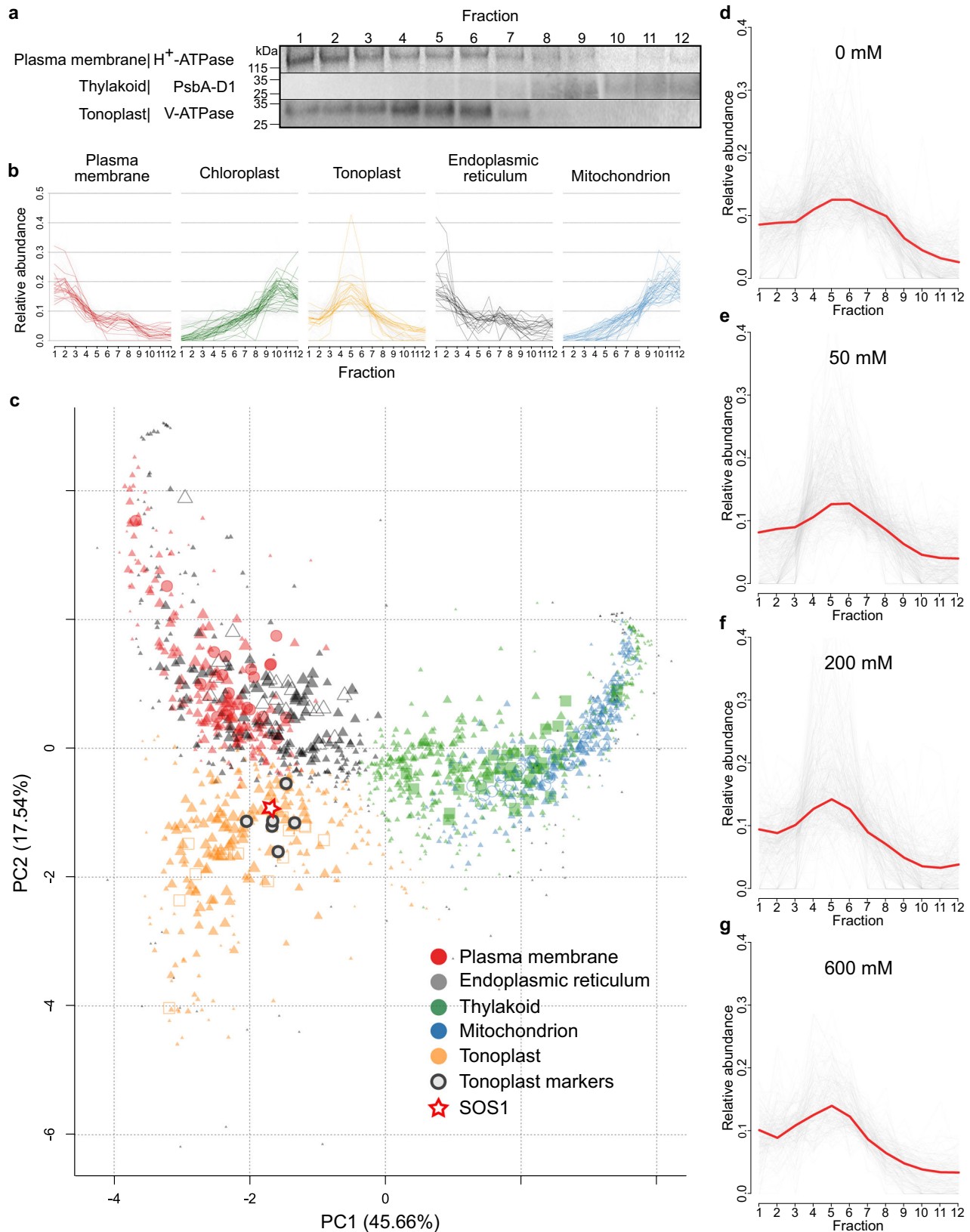

colocalization of eGFP and VAMP711 can be seen in Fig. 5m–o and Supplementary Fig. 12.

These results suggest the localization of SOS1 to the tonoplast in *S. bigelovii* and a high and constitutive abundance of SbiSOS1 in shoot cells. SbiSOS1 may have a role for rapid Na⁺ sequestration into the vacuole, enabling the cells to accumulate high Na⁺ concentrations.

Further inspection of SOS1 sequences revealed high conservation across plant species, with increased differences at the N and C termini and some variable regions at the middle of the protein (Supplementary Fig. 15). Similar results were obtained when analyzing the nonsynonymous (Ka) and synonymous (Ks) substitution rates[78] in *S. bigelovii* vs Arabidopsis *SOS1*. Ka/Ks greater than one were observed at the

**Fig. 4 | Spatial protein profiles and validation of marker abundance profiles generated with pRoloc from membrane preparations from shoots of *S. bigelovii* plants treated with 200 mM NaCl. a** Western blot analyses of organelle marker proteins for the plasma membrane (H⁺-ATPase), thylakoid (PsbA-D1), and tonoplast (V-ATPase) on 5 μg of proteins from each of the 12 density fractions. Representative image of 3 replicates. **b** Relative protein abundance profiles identified with pRoloc of organelle marker proteins on the 12 density fractions. **c** Principal component analysis of protein predicted subcellular localization by pRoloc. The size of the symbol represents confidence values of localization. Plasma membrane: red, markers as circles; endoplasmic reticulum: gray, markers as open triangles; chloroplast: green, markers as squares; mitochondrion: blue, markers as open circles; tonoplast: orange, markers as open squares; tonoplast marker proteins, encircled in black: Vacuolar ATPase subunits (**a**, **b**, **c** and **d**) Vacuolar pyrophosphatase, and Tonoplast intrinsic protein 1–3; and red star, SOS1. **d**–**g** Relative protein abundance profiles of SOS1 (red) and proteins allocated to the tonoplast (gray) at different treatments. **d** 0 mM NaCl. **e** 50 mM NaCl. **f** 200 mM NaCl. **g** 600 mM NaCl.

N-terminus and were particularly high at a region close to the C-terminus of the protein (Supplementary Fig. 16). It is possible that the difference in some of these residues contribute to the neo-localization of SbiSOS1 to the vacuole in *S. bigelovii*, yet further investigation is required.

## NaCl increases the abundance of membrane proteins related to ATP synthesis

We then looked at the protein responses upon salt treatment. Of the 2118 unique protein clusters, 687 were found to be differentially abundant in at least one of the treatments (Supplementary Fig. 17 and Supplementary Data 6). Hierarchical clustering of the differentially abundant proteins showed a clear two-group clustering between replicates from each treatment and between plants treated with 0 and 50 mM NaCl and those treated with 200 and 600 mM NaCl (Supplementary Fig. 17). GO enrichment analysis of proteins significantly more abundant in plants treated with 200 and 600 mM NaCl revealed enrichment in the terms 'cellular respiration', 'ATP synthesis', 'generation of precursor metabolites and energy', and 'cellular nitrogen compound biosynthetic process', among others (Supplementary Data 5). Plants grown in 0 mM and 50 mM were enriched in the terms 'programmed cell death', 'organelle fusion', 'response to stimulus', and 'calcium ion homeostasis' (Supplementary Data 5). While a direct comparison of the enrichment analyses between gene expression and protein abundances is difficult, as one comprises all gene responses and the other focuses on membrane protein abundances, similar responses were observed between both analyses (Fig. 3 and Supplementary Data 5). Plants grown in higher concentrations of NaCl were both enriched in terms related to energy production and growth, while plants grown without or at lower concentrations of NaCl were enriched in terms related to stress. Both results are in agreement with the beneficial effect of NaCl in *Salicornia* growth and development. Other terms such as 'response to stress', 'response to nitrogen transport', and 'response to water deprivation' were uniquely found in the gene expression analysis in plants grown in the absence of NaCl. Closer inspection revealed that these terms were also firstly identified in the proteomics enrichments prior to multiple testing correction but were no longer significant after *p*-value adjustment (Supplementary Data 5). This could be the consequence of working with a reduced subset of proteins, reducing the statistical power, but could also be a reflection of the differences in responses between soluble and membrane proteins, or the difference between mRNA turnover rates and protein abundances. MapMan pathway enrichment analysis produced similar results, suggesting increased signaling and vesicle trafficking in plants grown in 0 mM and 50 mM NaCl and an increase in ribosome biogenesis and ATP synthesis in plants treated with 200 and 600 mM NaCl (Supplementary Tables 5 and 6).

## Identification of salt tolerance related genes in yeast

To identify genes of *S. bigelovii* involved in salt tolerance, genes were selected for salt tolerance assays in the salt sensitive yeast strain AXT3 (Δena1-4::HIS3 Δnha1::LEU2 Δnhx1::TRP1)[79] based on the following criteria: (1) Genes which were differentially expressed in response to NaCl; (2) Genes previously associated with salinity tolerance; (3) Genes with a domain or description associated with Na, K, Cl, or Ca; and (4)

Preference was given to transmembrane proteins predicted to have at least six transmembrane helices, as this would advocate for transporters and channels. Based on these criteria, 45 genes were cloned and 11 showed a phenotype in response to NaCl (Fig. 6a, b, Supplementary Fig. 18, Supplementary Table 7, Supplementary Data 7). *SbiSOS1* was toxic to *E. coli* and we were not able to retrieve a functional transformant for its further analysis in yeast.

Increased salinity tolerance was observed for four genes: *SbiKAT2*, *SbiNHX4*, *SbiOPT4*, and *SbiSALTY* (Fig. 6a and Supplementary Fig. 18). *SbiKAT2* is similar to Arabidopsis potassium channel *KAT2*. It was downregulated upon NaCl treatment, it greatly increased yeast salt tolerance, and its activity as a K⁺ channel was confirmed when grown in high concentrations of K⁺, leading to K⁺ toxicity and cell death (Supplementary Fig. 19). *NHX* genes showed different phenotypes in yeast but were not differentially expressed in *S. bigelovii* upon NaCl treatment. *SbiNHX4* greatly increased yeast salt tolerance and is similar to Arabidopsis *NHX4*, belonging to the Class I exchangers (NHX1-NHX4) that are localized to the tonoplast[44–46] and are considered to be primarily responsible for the compartmentalization of K⁺ in the vacuole[43,80–82]. SbiOPT4 is similar to the oligopeptide transporter OPT4 in Arabidopsis, which has been described as a proton/oligopeptide transporter[83]. *SbiOPT4* expression increased upon NaCl treatment and conferred a moderate increase in salinity tolerance in yeast, particularly when grown in AP medium (Supplementary Fig. 18). The most pronounced phenotype was observed with an RGG protein, which we called SALTY. Its gene expression and protein abundance increased upon NaCl treatment, and it localized to the cytosol in yeast (Supplementary Figs. 11 and 20). RGG proteins are not well studied in plants, but they are the second largest RNA binding protein family in humans[84]. Sequence analysis of SbiSALTY suggested it to be an intrinsically disordered protein (IDP), a protein that does not have a defined three-dimensional structure. The overexpression of Arabidopsis homologs AtRGGA (AT4G16830) and the RGG protein (AT4G17520) also increased salt tolerance in yeast, albeit to a lesser extent (Fig. 6a, c).

Reduced salt tolerance in yeast was observed for seven genes: *SbiHKT1*, *SbiNHX1*, *SbiNHX2*, *SbiENT1*, *SbiSLC5-6*, *SbiPQ*, and *SbiIDP1* (Fig. 6b and Supplementary Fig. 18). Expression of *SbiHKT1* decreased with NaCl treatment (Fig. 3e) and its induction in yeast led to cell death. HKT1;1 has been reported to be involved in root Na⁺ accumulation and Na⁺ retrieval from the xylem, preventing Na⁺ accumulation in leaves of Arabidopsis[52,53]. Two *NHX* genes (*SbiNHX1* and *SbiNHX2*) decreased salt tolerance in yeast, contrarily to what was observed for *SbiNHX4*. This may be due to differences in their subcellular localization in yeast, with SbiNHX1 and SbiNHX2 primarily localizing to lysosomes and SbiNHX4 to the vacuole and ER (Supplementary Fig. 21). *SbiENT1* is similar to Arabidopsis equilibrative nucleoside transporter 1 (ENT1). It was predicted to be localized to the tonoplast and was upregulated upon NaCl treatment. Equilibrative nucleoside transporters mediate the transport of purine and pyrimidine nucleosides[85]. In Arabidopsis, ENT1 is localized to the tonoplast and its overexpression leads to stunted growth[86]. *SbiSLC5-6* expression did not change upon NaCl treatment and was predicted to localize to the tonoplast. It resembles a sodium-coupled amino acid transporter and has an SLC5-6-like_sbd superfamily domain. SLC5 proteins cotransport Na⁺ and a substrate, usually

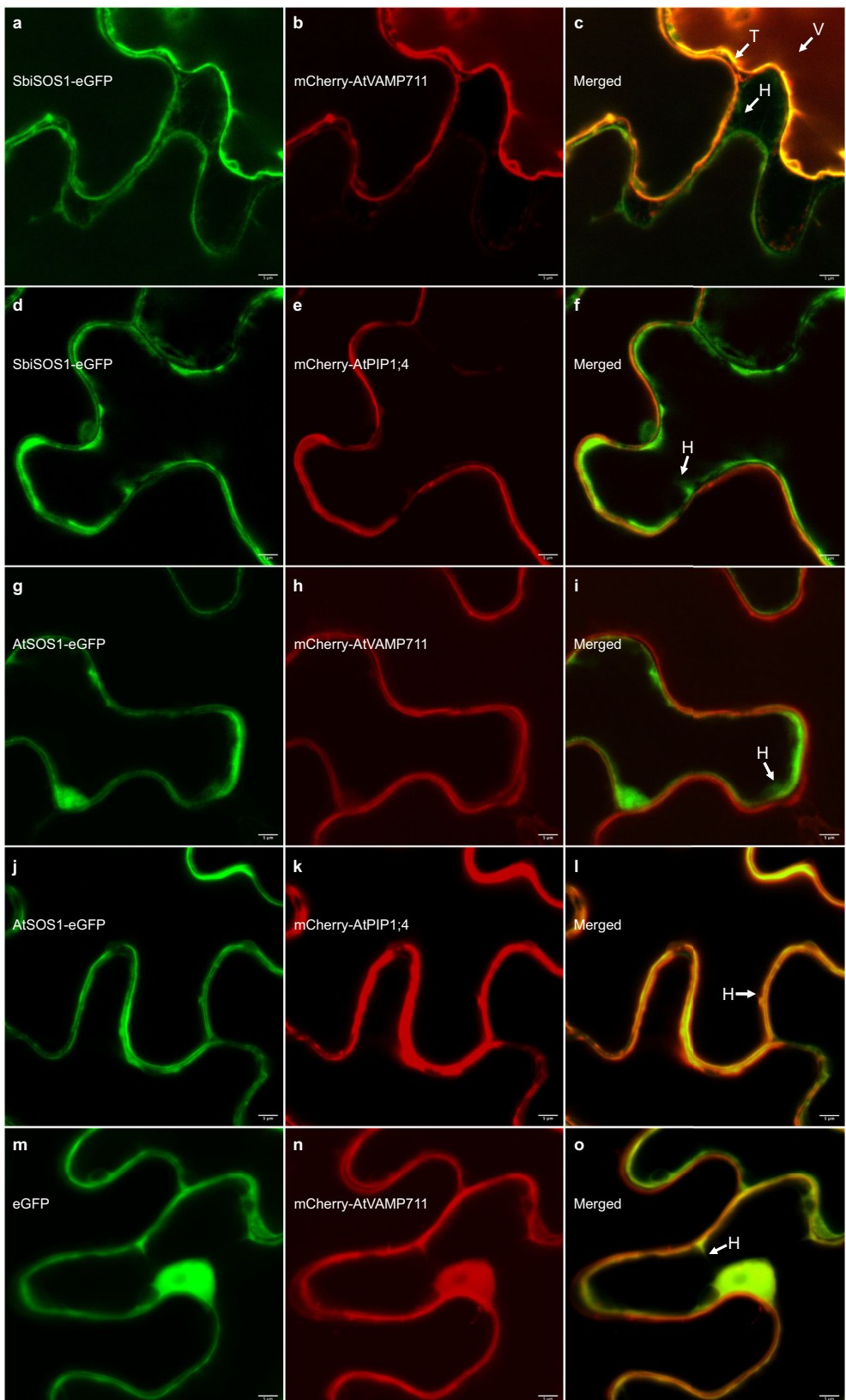

**Fig. 5 | Subcellular localization assays of *S. bigelovii* and Arabidopsis SOS1 in tobacco leaves. a–f**, SbiSOS1 subcellular localization assays. **a** SbiSOS1-eGFP. **b** mCherry-AtVAMP711 (vacuolar marker). **c**, Merged image. **d** SbiSOS1-eGFP. **e** mCherry-AtPIP1;4 (plasma membrane marker). **f** Merged image. **g–l** AtSOS1 subcellular localization assays. **g** AtSOS1-eGFP. **h** mCherry-AtVAMP711. **i** Merged image. **j** SbiSOS1-eGFP. **k** mCherry-AtPIP1;4. **l** Merged image. **m–o** eGFP control. **m** eGFP. **n** mCherry-AtVAMP711. **o** Merged image. Cells were plasmolyzed with 300 mM mannitol to aid in the visualization of the plasma membrane and tonoplast. T Tonoplast, V Vacuole, H Hechtian strands. All scale bars at 5 μm. Representative images of two independent plant inoculations per construct.

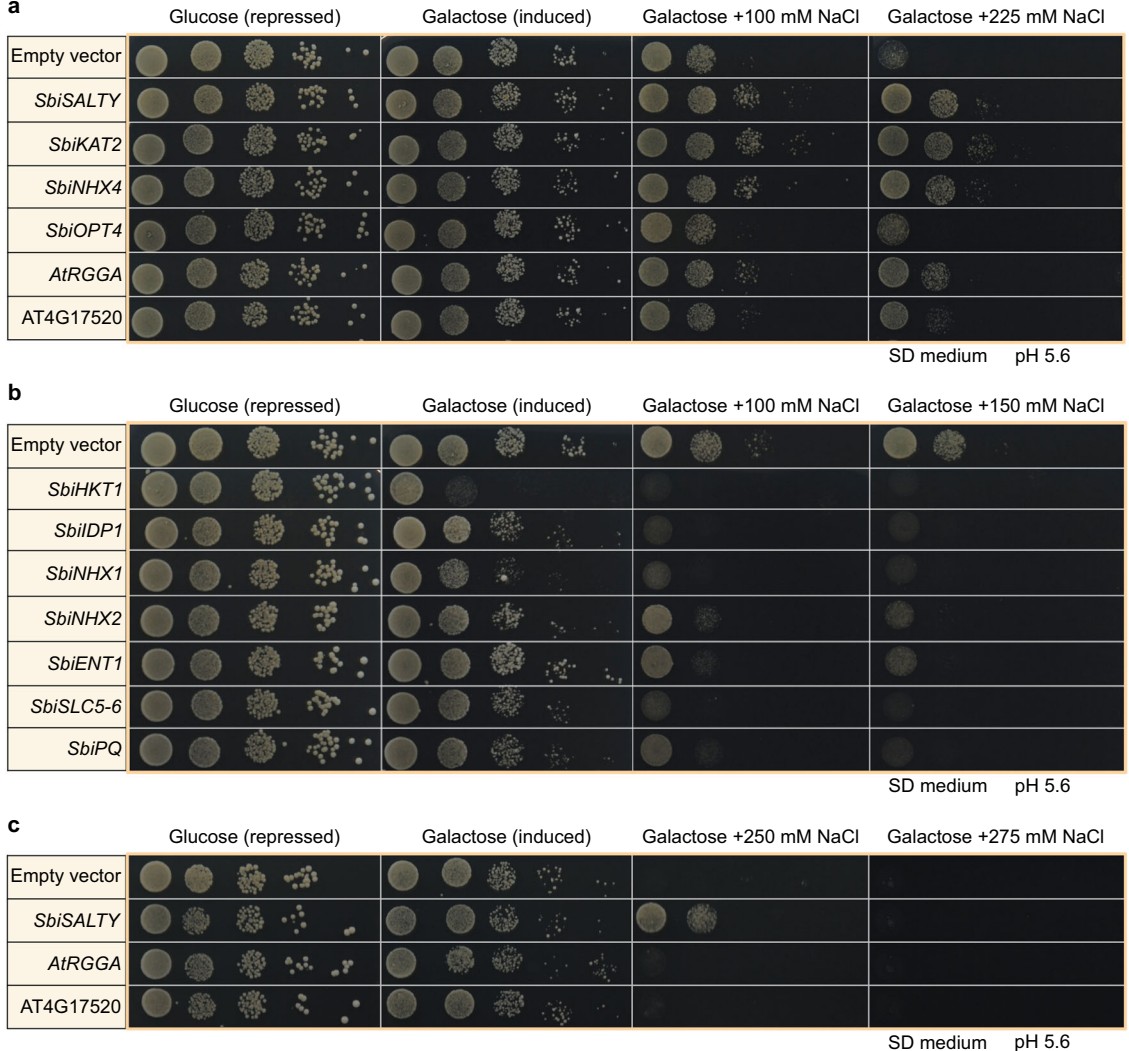

**Fig. 6 | Yeast spot assays in AXT3 strain at different NaCl treatments. a** genes increasing salt tolerance. **b** genes decreasing salt tolerance. **c** RGG proteins at high NaCl. Assays were done in SD medium pH 5.6 and supplemented with 2% galactose for gene induction. Representative images of three replicates.

glucose[87], whereas SLC6 proteins utilize Na⁺ and Cl⁻ to cotransport substrates[88]. *SbiPQ* expression did not change upon NaCl treatment and it is similar to the PQ-loop repeat family of proteins. PQ-loop proteins are not well studied in plants, but have recently been shown to function as H⁺/monovalent cation transporters in plants[89] and as H⁺/ amino acid transporters in yeast[90] and human[91,92]. *SbiIDP1* was upregulated upon NaCl treatment and greatly induced yeast cell death, particularly upon NaCl treatment. It had no significant similarity to any Arabidopsis protein, and similarly to SbiSALTY, it is predicted to be an intrinsically disordered protein, to form coiled coils, and was localized to the cytosol in yeast (Supplementary Fig. 22).

## The RGG protein SALTY is an intrinsically disordered protein and is localized to the ER

We further characterized SbiSALTY, as it greatly increased salt tolerance in yeast. Sequence analyses suggests it to be an intrinsically disordered protein (Fig. 7a, Supplementary Figs. 23 and 24) and be able to form coiled-coils (Fig. 7b), which could allow for oligomerization. To confirm its structure, SbiSALTY was expressed and purified in *E. coli* for circular dichroism (CD) and NMR analysis. The CD spectrum showed that SbiSALTY is predominantly disordered, having a minimum around 200 nm and very low ellipticity above 210 nm[93,94], missing both well-pronounced classical secondary

structures and distinct tertiary structural motifs (Fig. 7c). Temperature-based CD analyses at three different wavelengths, 201, 208, and 222 nm, showed no significant alterations at temperatures ranging from 20 to 92 °C (Fig. 7d), supporting the notion that SbiSALTY is an IDP. To explore the detailed structure of SbiSALTY, we expressed and purified the uniformly ¹⁵N-labeled protein for NMR analysis. The acquired high-resolution 2D ¹H-¹⁵N TROSY-HSQC spectrum indicates that SbiSALTY is an IDP as the signal dispersion is minimal, particularly in the ¹H dimension (Fig. 7e). This minimal dispersion is typical of IDPs[95,96]. The overall number of identified ¹H/¹⁵N backbone amide correlations found in the spectrum fits the expected amount based on the SbiSALTY primary structure, confirming the one dominant conformation of the disordered state of the intact protein in solution. We did not observe any ¹H/¹⁵N chemical shift changes with concentrations ranging from 50 μM to 1 mM. These data suggest that SbiSALTY is water-soluble and intrinsically disordered.

Since SbiSALTY is a water-soluble protein, its abundance in our membrane proteomes was highly diminished and its allocation to a subcellular compartment should be taken with caution. Nevertheless, SbiSALTY was detected and was most abundant at higher NaCl treatments and in proximity to ribosomal proteins (Supplementary Fig. 11). Expression of *SbiSALTY* in rice protoplasts suggests its localization to

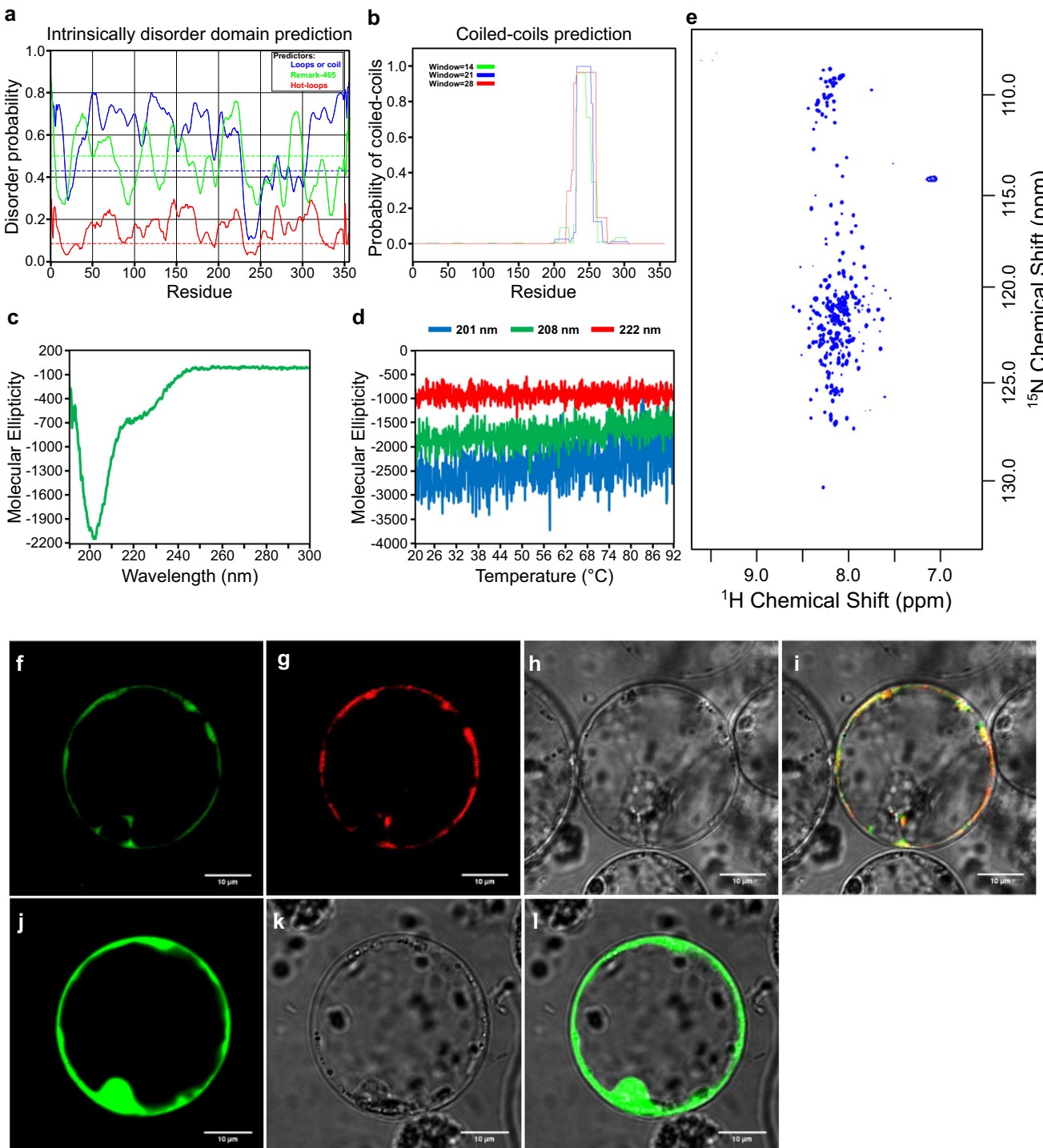

**Fig. 7 | Structure prediction for SbiSALTY and its subcellular localization in rice protoplasts. a** Intrinsically disordered domain prediction. Dotted lines represent random expectation values for each model. **b** Coiled-coils prediction. **c** CD measurements of SbiSALTY at 30°C. Wavelength range from 190 nm to 260 nm showing secondary structure and wavelength range from 250 nm to 300 nm showing tryptophan and tertiary structure. **d**, Temperature melting curve analysis of 201 nm, 208 nm and 222 nm of SbiSALTY in CD ranging from 20 – 92 °C. **e** [1]H-[15]N HSQC spectrum of SALTY-WT in 800 MHz NMR analysis**. f–l** Protein expression in rice protoplasts. Representative images of three independent transfections per construct. **f** SbiSALTY-EGFP. **g** ER-marker ER-rk (AtWAK2 signal peptide-mCherry-ER retention signal)[187]. **h** Bright field. **i** Merged image. **j** empty vector expressing EGFP. **k** Bright field. **l** Merged image.

the ER, as it co-localized with an ER marker (Fig. 7f–l). The homolog of this gene in Arabidopsis (*RGGA*, AT4G16830) was observed to localize to the cytosol and interact with non poly-A RNA[97], suggesting it might interact with ribosomes. Ribosomes bound to the ER (rough ER) cause an increase in the membrane's density when compared to the ribosomes-devoid smooth ER, leading to an increased specific gravity

between 1.15–1.18[98,99], where SbiSALTY was most abundant. Most ribosomal proteins were also identified across these fractions. Sequence analysis of SbiSALTY with Phyre2[100] predicted it to be a highly disordered protein, but showed a short conserved not disordered region to be most similar to the 40 S ribosomal protein S7 (Supplementary Fig. 24). This small, conserved region falls within the

region predicted to form coiled-coils and could help SbiSALTY interact with ribosomes in the ER of plants.

## Discussion

Here we studied the responses of the highly salt tolerant *S. bigelovii* to NaCl and generated the first genomic resources for the genus *Salicornia*. These plants are able to accumulate very high concentrations of NaCl in their shoot whilst avoiding visible symptoms of ion toxicity. Nevertheless, *S. bigelovii* plants displayed enhanced growth upon the addition of NaCl and accumulated very high concentrations of $Na^+$ and $Cl^-$ in its shoots and roots (Fig. 1).

The positive effect of NaCl on growth was also observed at a transcriptional and protein level, where genes and proteins related to growth, photosynthesis, cell wall modification, water transport, and ATP synthesis were found to be upregulated upon NaCl treatment. The high concentration of both $Na^+$ and $Cl^-$ in shoots and its positive correlation with water content suggest a possible role for both ions as osmolytes. $Na^+$ was found to be more effective than $Cl^-$ or $K^+$ for supporting *S. europaea* growth[101]. Nevertheless, a set of adaptations and possibly a tight regulation of specialized ion transporters would be required to avoid ion toxicity, moving ions out of the cytosol, and in particular into the large volume of the vacuole. We did not observe many differences at a transcriptional or protein level between plants treated with 200 mM and 600 mM NaCl for 6 weeks. It is possible that plants had already acclimated to the increased salinity after 6 weeks, or that post-translational modifications such as phosphorylation play an important role in the modulation of these responses through the regulation of transporters. It would be interesting to assess the prevalence and impact of these modifications.

Gene ontology enrichment analysis of unique orthogroups in *Salicornia* revealed enrichment in terms related to responses to high light intensity, toxin metabolism and ion channel activity, in the form of calcium channels. These calcium channels could help in calcium signaling upon NaCl stress, allowing for a fast response. However, neither an enrichment specific to $Na^+$ transport nor an expansion in known $Na^+$ transporters were observed. Suggesting that *Salicornia* might use the same set of genes found in other plants, albeit adapted for highly saline environments, or possess genes involved in salinity tolerance that have not yet been discovered.

Analysis of the organellar membrane proteomes revealed a possible neo-localization of SOS1 to the tonoplast (Fig. 4). SbiSOS1 was highly abundant in all treatments and was always allocated with high confidence to the tonoplast in our organellar membrane proteomes. Transient expression of *SbiSOS1* in tobacco cells confirmed its presence in the tonoplast (Fig. 5). SbiSOS1 also localized to the plasma membrane, raising the question if this dual localization of SbiSOS1 also occurs in *Salicornia* or if *Salicornia* possesses a mechanism preventing SbiSOS1 accumulation in the plasma membrane. However, this could also be the result of excess protein being targeted to the plasma membrane due to overexpression with the 35 S promoter, as the secretory pathway is the default pathway for membrane protein localization[77]. In silico analysis of SbiSOS1 revealed regions of increased divergence when compared to Arabidopsis SOS1, particularly at the N and C termini of the protein, which could be involved in its localization to the tonoplast. Deletion of the C-terminal of Arabidopsis SOS1 results in its loss of autoinhibition[102], further investigation is required to see if the observed differences in this region lead to changes in regulation or activity. SbiSOS1 has been described as a $Na^+/H^+$ antiporter localized to the plasma membrane in Arabidopsis, where it transports $Na^+$ outside of the cell[48,71–74]. The novel localization of SOS1 to the tonoplast in *S. bigelovii* could allow for high concentrations of $Na^+$ to be compartmentalized into the vacuole, preventing $Na^+$ from causing toxicity in the cytoplasm, and allowing its use as an osmolyte for water retention. Compartmentalization of $Na^+$ in the vacuole would be advantageous in highly saline environments, as the otherwise

continuous $Na^+$ extrusion from the cytoplasm to the extracellular space would be highly energetically demanding. V-ATPases, vacuolar pyrophosphatases, and TIPs were amongst the most abundant proteins in our membrane proteomes. The V-ATPase and vacuolar pyrophosphatases would generate $H^+$ ions to be used by the $Na^+/H^+$ antiporter activity of SOS1[103], while the TIPs would allow water flow into the vacuole, helping the increase in volume of *Salicornia* vacuoles that occurs in response to elevated salinity.

We identified 11 proteins that showed a phenotype in yeast – four increased salt tolerance and seven decreased it. Two NHX proteins decreased salt tolerance (SbiNHX1 and SbiNHX2), while one greatly increased it (SbiNHX4), which could be related to their different subcellular localization in yeast. Alternatively, their predicted C-terminus localization with respect to the membrane was opposite in SbiNHX4 when compared to SbiNHX1 and SbiNHX2 and could play a role in their regulation, as the binding of proteins to the C-terminus can influence their activities[104,105]. We also identified an *SbiHKT1* whose expression was reduced *in planta* upon NaCl treatment, and which led to cell death in yeast. *In planta*, HKT1 seems likely to reduce root-to-shoot translocation of $Na^+$ by removing $Na^+$ from the transpiration stream into xylem parenchyma cells[106]. The detrimental effect that some genes had in yeast might not only be attributed to a negative effect of the expression of these genes upon salinity in plants but could also be a consequence of subcellular mis-localization in yeast cells, as expression in yeast may not accurately reflect organellar localization *in planta*. Furthermore, it could also be a consequence of the differences between the unicellular nature of yeast as opposed to the multicellularity of plants, as different plant tissues might behave differently in the presence and compartmentalization of NaCl.

Two predicted intrinsically disordered proteins, SbiIDP1 and SbiSALTY, showed contrasting phenotypes in yeast. SbiSALTY is an Arginine-Glycine-Glycine (RGG) motif protein and greatly increased salt tolerance in yeast. RGG proteins are not well studied in plants, but in humans are the second largest RNA binding protein family, and have been associated with transcription, pre-mRNA splicing, RNA transport, translation, ribonucleotide biogenesis, and other mechanisms such as DNA repair and modulation of chromatin structure[84,107–109]. In Arabidopsis, the overexpression of *AtRGGA*, encoding an RGG protein, increased salt and drought tolerance compared to wild type plants[97]. The protein localized to the cytosol, and bound non poly-A RNA[97]. Expression of Arabidopsis homologs *RGGA* and *AT4G17520* in yeast also increased salt tolerance, albeit to a lesser extent than *SbiSALTY* (Fig. 6). NMR and CD analyses confirmed SbiSALTY to be an intrinsically disordered protein. Some of the most commonly known IDPs are the Late Embryogenesis Abundant (LEA) proteins, which are involved in dehydration tolerance[110–112]. IDPs are considered to be primary constituents of membrane-less compartments within the cell[113–115]. Membrane-less compartments display liquid-like properties[116] and form structures such as stress granules and p-bodies, which have been observed to contain ribosomal subunits and translation initiation factors[117–119]. The RGG domain is intrinsically disordered, and RGG proteins have been associated with the formation of membrane-less compartments such as RNA-granules in *Drosophila*, *C. elegans*, and humans, regulating RNA stability and translation[84,120,121]. In our membrane proteomes, SbiSALTY was found to be most abundant at densities where ribosomal proteins are found and the expected density of the rough ER. However, a precise localization of SbiSALTY in our proteomes is difficult, as soluble protein abundances were reduced during membrane enrichment and no $Mg^{2+}$ was added to our density fractions to preserve the ribosomes binding to the ER[98,99]. Nevertheless, *in planta*, SbiSALTY was found to be localized to the ER. It was recently found that three Arabidopsis RGG proteins, AtRGGA (AT4G16830), AtRGGB (AT4G17520), and AtRGGC (AT5G47210), interact with structural components of the 40 S and 60 S ribosomal subunits[122]. SbiSALTY has a high degree of similarity with these three

proteins, 62%, 55%, and 58%, respectively and is likely to also interact with ribosomal proteins. The intrinsically disordered nature of SbiSALTY, together with its predicted ability to form coiled-coils at a region highly similar to the 40 S ribosomal proteins S7, could provide it with the required flexibility to form part of membrane-less compartments, where it could bind to RNA-containing structures such as ribosomes, possibly stabilizing or protecting them during salt stress. The increased gene expression and protein abundance of SbiSALTY upon salt stress suggests a beneficial effect of this protein upon highly saline conditions, posing the question of why this protein is not always expressed at the same levels. It could be that SbiSALTY has a "penalty", perhaps its binding to ribosomes limits translation under non-saline conditions yet provides a protecting effect at higher salinities, allowing for sustained translation.

The analyses presented in this study provide new insights into the responses of *S. bigelovii* to salinity and the possible adaptations and mechanisms that allow it to thrive in highly saline environments. The resources generated open the possibility for further research in *Salicornia*, which will increase our knowledge of salinity tolerance and will help in the development of salt tolerant crops.

## Methods

### Plant material and growth conditions

*Salicornia bigelovii* seeds were kindly provided by Dr. E. Glenn of the Environmental Research Laboratory, University of Arizona, Tucson, USA. *Salicornia europaea* seeds were collected in the Dead Sea area and were kindly provided by Dr. Moshe Sagi of the Blaustein Institute for Desert Research (BIDR), Israel. For phenotypic, transcriptomic, and proteomic experiments in *S. bigelovii*, seeds derived from inbred populations were sown in Metro-Mix® 360 soil (SunGro Horticulture, USA) contained in plastic trays and watered daily to 70% soil water holding capacity. *S. bigelovii* plants of one inbred line were grown during the months of May–July 2015 in a glasshouse in KAUST under natural irradiance and kept at a constant temperature of 28/24 °C day/night with 65% relative humidity. Five weeks after sowing, *S. bigelovii* plants were treated with saline water to reach a final concentration of 0, 5, 50, 200, or 600 mM NaCl in the soil. Three trays per treatment, for a total of 12 trays, were grown for the experiments with all replicates showing consistent phenotypes.

### Plant phenotype and ion content

Shoot diameter imaging and measurements were obtained by finely slicing *S. bigelovii* shoots with a surgical blade and examining them under a light microscope (AZ100 Multizoom, Nikon, Japan). Shoot area cross sections were measured with NIS-Elements BR software (Nikon). For ion content, shoots and roots of *S. bigelovii* were harvested 1 and 6 weeks after salt treatment. Samples were rinsed with MilliQ water and had the surface water removed with paper towels prior to fresh mass measurements. Samples were then dried in an oven at 70 °C for 3 days and had their dry mass measured. 10 mg of dried tissue were mixed with 1.5 mL of 500 mM nitric acid, heated at 80 °C for 3 h, and used for Na and K content analyses by flame photometry (Flame photometer model 420, Sherwood Scientific Ltd, United Kingdom) and Cl⁻ content (Chloride analyser model 926, Sherwood Scientific Ltd, United Kingdom). Flame photometry measures the total element in a sample, not distinguishing between ionic and non-ionic forms; nevertheless, these measurements will be referred to as "Na⁺" and "K⁺" and not as "Na" and "K" as most of these elements are found as ions in the cell. In contrast, the choride analyzer uses titration with silver nitrate to measure chloride in its ionic form, "Cl⁻". Water content was calculated as (fresh mass – dry mass) / dry mass. Shoot sap osmolality was obtained by crushing *S. bigelovii* shoots and centrifuging at 10,000 x *g* for 2 min. 10 μL of the supernatant was measured with a VAPRO® (ELITechGroup). For mean comparison across treatments, statistical analyses of three samples per treatment were done with the R package stats v.3.6.3[123] in

### DNA extraction, library preparation, and sequencing

High molecular weight (HMW) DNA was isolated from the young secondary stems of a single plant after dark-treatment for 48 h. DNA was isolated using a modified CTAB method. Briefly, 5 grams of tissue were combined with 10 mL of CTAB Buffer (2% (w/v) CTAB, 1.4 M NaCl, 100 mM Tris-HCl (pH 8), 20 mM EDTA (pH 8), 2% (w/v) PVP (MW 10,000), 1% (v/v) Triton X-100, 0.1% (v/v) βME, 1.9 mg/mL RNase-A, and 10 mg/mL proteinase K). Extracts were incubated and mixed for 45 min at 65 °C before isolation of the aqueous phase with chloroform:isoamyl alcohol (24:1), done thrice. The DNA was precipitated in 0.6 volumes of pure isopropyl alcohol. The HMW DNA was pelleted and washed with 70% (v/v) ethanol and resuspended in TE buffer (pH 8.0). HMW DNA was further purified using AMPure PB beads (Beckman Coulter, USA) and resuspended in TE buffer (pH 8.0). High molecular weight DNA was used for HiFi sequencing of 4 PacBio HiFi SMRT cells, accounting for 57,386,832,994 bp or ~28 x coverage for *S. bigelovii* and 2 PacBio HiFi SMRT cells, accounting for 17,950,298,972 bp or ~34 x coverage for *S. europaea* and were sequenced in a PacBio Sequel II system at The Arizona Genomics Institute of The University of Arizona (USA). For genome size estimation using *k*-mer analysis, Illumina per species were sequenced in a NovaSeq 6000 system at The Arizona Genomics Institute of The University of Arizona (USA), resulting in 112,237,052,136 bp or ~55x coverage for *S. bigelovii* and 29,229,294,386 bp or ~56x coverage for *S. europaea*.

### RNA extraction and library preparation

To generate reference transcriptomes for gene annotation, RNA samples from shoots and roots from *S. bigelovii* and *S. europaea* were extracted with an RNeasy Plant Mini Kit (Qiagen). RNA quality was determined with a Bioanalyzer 2100 using an RNA 600 Nano kit (Agilent, USA). High quality of RNA was used to prepare RNA-Seq libraries with NEBNext® Ultra™ Directional RNA Library Prep Kit for Illumina® (New England BioLabs, USA) according to the manufacturer's instructions. RNA-Seq libraries were sequenced in a NovaSeq 6000 system (Illumina, USA) under paired-end mode and 151 bp read length at the Bioscience Core Labs (KAUST), resulting in 552 million (~83 Gb) and 488 million (~73 Gb) reads for *S. bigelovii* and *S. europaea*, respectively.

For gene differential expression analysis of shoots for *S. bigelovii* plants treated with 0, 50, 200, or 600 mM NaCl for 1 and 6 weeks, RNA was extracted with TRIzol™ and an RNA MiniPrep kit (Zymo Research, USA), following the manufacturer's protocol. RNA quality was determined with a Bioanalyzer 2100 using an RNA 600 Nano kit (Agilent, USA) using only RNA with RIN values greater to 8. Libraries for RNA-Seq were prepared with NEBNext® Ultra™ Directional RNA Library Prep Kit for Illumina® (New England BioLabs, USA) for three replicates per treatment resulting in 24 libraries with a mean insert size of 300 bp. DNA quality was determined with a high sensitivity DNA kit (Agilent, USA) using a Bioanalyzer 2100. Libraries were barcoded and sequenced in two lanes of Illumina HiSeq 2000 generating ~540 million paired end reads of 2 × 101 bp or ~22.5 million reads per library (Supplementary Data 4) at the Bioscience Core Labs (KAUST, Saudi Arabia).

### Chromosome counting and genome size estimation

To count the number of chromosomes of *S. bigelovii* and *S. europaea*, chromosome spreads were prepared from actively growing roots as previously described[124] with minor modifications. Roots were collected into 50 mM phosphate buffer (pH 7.0) containing 0.2% (v/v) β-mercaptoethanol. The roots were transferred to 0.05% (w/v) 8-hydroxyquinoline and incubated for 3 h at room temperature, fixed in 3:1 ethanol:acetic acid fixative and stored in 70% (v/v) ethanol at −20 °C for further use. At least twenty roots were washed three times with distilled water within 10 min and once in 1xKCl buffer (pH 4.0) for 5 min. Meristematic zones of the roots were excised and digested in

200 μl of the enzyme solution containing 4% (w/v) cellulase and 1% (v/v) pectolyase for 48 min at 37 °C, then 200 μL of TE buffer (pH 7.6) was added to stop the reaction. Root partitions were washed three times in 96% (v/v) ethanol and 25 μL of 9:1 ice-cold acetic acid: methanol mixture was added. The root tips were broken with a rounded dissecting needle and the tube was kept on ice for 5 min before dropping. The suspension (5 μL) was dropped onto microscopic slide placed in a humid polystyrene box. The preparations were air-dried and counterstained with DAPI mounted in VECTASHIELD Antifade Mounting Medium (Vector Laboratories, Burlingame, CA, USA). The slides were examined with an Axio Imager Z.2 Zeiss microscope (Zeiss, Oberkochen, Germany). Pictures were captured by the ISIS software 5.4.7 (Metasystems) and chromosomes were counted manually. A minimum of five mitotic metaphase plates per slide were captured. Genome sizes were estimated through flow cytometry and *k*-mer frequencies. Nuclear DNA content was estimated by flow cytometry from three different individual inbred plants per species, as previously described[125], using *Zea mays* L. 'CE-777 (2 C = 5.43 pg DNA) and *Solanum lycopersicum* L. 'Stupické polní rané' (2 C = 1.96 pg DNA) as internal reference standards for *S. bigelovii* and *S. europaea*, respectively. Samples were analyzed using a CyFlow Space flow cytometer (Sysmex Partec GmbH, Görlitz, Germany) equipped with a 532 nm laser. The gain of the instrument was adjusted so that the peak representing G1 nuclei of the standard was positioned approximately on channel 100 (*Zea mays*) or 170 (*Solanum lycopersicum*) on a histogram of relative fluorescence intensity when using a 512-channel scale. Three individual plants were sampled and analyzed three times. At least 5000 nuclei per sample were analyzed and 2 C nuclear DNA contents were calculated from means of G1 peak positions by applying the formula: 2 C nuclear DNA content = (sample G1 peak mean) × (standard 2 C DNA content) / (standard G1 peak mean). The mean nuclear DNA content in pg was converted to genome size in bp using the conversion factor 1 pg DNA = 0.978 Gbp[125]. Genome size estimation through *k*-mer frequencies was done with FindGSE[28] by counting *k*-mer frequencies of Illumina or HiFi reads with Jellyfish[126]. For genome size estimation with Illumina reads, reads were first trimmed with Trimmomatic[127] and quality checked with FastQC[128] prior to *k*-mer analysis.

## Genome assembly and annotation

The genomes of *S. bigelovii* and *S. europaea* were assembled using the PacBio HiFi reads with HiFiasm[129]. HiFiasm was run with the --n-hap 4 and -s 0.35 commands for *S. bigelovii* and with the -s 0.35 command for *S. europaea*. The genome assembly completeness was measured with Merqury[29], resulting in 97.13% and 99.29% read *k*-mer presence in the genome assemblies for *S. bigelovii* and *S. europaea*, respectively. The genomes of the *Salicornia* species were masked for repetitive elements by generating a de novo repeat library with RepeatModeler for each species and used to mask the genome with RepeatMasker[130–132]. Hints for genome annotation were generated by mapping RNA-Seq data from each species into their corresponding genome with HISAT2[133]. Protein predictions on the masked genomes were carried out with BRAKER2[134–141], using the mapped RNA-Seq reads and the Viridiplantae dataset from OrthoDB[142] as hints. Predicted proteins were then annotated with matches against Swiss-Prot, TrEMBL[31], NCBI NR[32] with BLASTp[143,144], and the Kyoto Encyclopedia of Genes and Genomes (KEGG)[35] with KofamKOALA[145], while protein domains were identified with InterProScan[146]. Gene ontology terms[38,39] were retrieved from the Swiss-Prot annotation and InterProScan results. MapMan[33,34] terms were retrieved with Mercator[34,147]. The completeness of the annotations was assessed with BUSCO using the Eudicotyledons v.10 dataset. Genome annotations can be found in Supplementary Data 1 and 2.

## Orthology and phylogenetic analyses

Groups of orthologous proteins were identified with OrthoMCL[37] for representative genomes of Brassicaceae: *Arabidopsis thaliana*

(Araport11)[148,149] and *Brassica rapa* (FPSc v1.3)[150]; Amaranthaceae: *Beta vulgaris* (v1.2)[151], *Chenopodium quinoa* (v1)[152], *Salicornia bigelovii* (this study), *Salicornia europaea* (this study), and *Spinacia oleracea* (v1)[153]; Fabaceae: *Glycine max* (Wm82.a2.v1)[154] and *Phaseolus vulgaris* (v2.1)[155]; Solanaceae: *Solanum lycopersicum* (ITAG4.0)[156] and *Solanum melongena* (v3)[157]; and Poaceae *Sorghum bicolor* (v3.1.1)[158]. One isoform per gene was used for the analysis. All genome protein predictions were re-annotated by performing BLASTp[143,144] against the Swiss-Prot database[31]. Gene Ontology (GO) terms[38,39] were extracted from each best hit for each protein. Orthogroups shared across lists were identified with the Venn diagram creator of the University of Gent (http://bioinformatics.psb.ugent.be/webtools/Venn/). GO enrichment analyses on the orthogroup lists were analyzed with BINGO[159,160] using a Benjamini and Hochberg false discovery rate correction[161] of 0.05.

Family phylogenetic reconstruction was done by selecting BUSCO orthologs from the Eudicotyledons v10 dataset present in every species, resulting in a set of 1,377 proteins. Protein sequences were aligned with MUSCLE[162,163], informative sites were identified and retained with Gblocks[164], the alignments were concatenated and an appropriate protein evolutionary model was identified with ModelTest-NG[165]. Phylogenetic analysis was done with maximum likelihood inference with RAxML-NG[166] under a JTT + I + G4 model and 1000 bootstrap replicates under the Transfer Bootstrap Expectation (TBE) method[167]. For the phylogenetic reconstruction of NHX proteins, Arabidopsis NHX protein sequences were searched in the genomes of all other species by BLAST, with a cutoff e-value of e-10. Only proteins longer than 200 aa were used for the analysis (Supplementary Data 4). Protein sequences were aligned with MUSCLE and an appropriate protein evolutionary model was identified with ModelTest-NG. Phylogenetic analysis was done with maximum likelihood inference with RAxML-NG under a JTT + G4 model and 1000 bootstrap replicates under the Transfer Bootstrap Expectation (TBE) method. Trees were visualized with iTOL[168]. For SOS1 sequence analyses, SOS1 proteins were aligned with MUSCLE and visualized with Jalview[169]. Nonsynonymous and synonymous substitution rates between *S. biglovii* and Arabidopsis *SOS1* were identified with KaKs calculator 3.0[78], dividing the coding sequence in 6 bp sliding windows of 57 bp in length.

## Gene differential expression analysis

For gene differential expression analysis of *S. bigelovii* shoots, RNA-Seq reads were adapter trimmed with Trimmomatic v0.39[127] and quality checked with FastQC[128]. Transcript abundances were estimated with Salmon[170] and differentially expressed genes were identified with DESeq2[171] with a minimum of 10 counts and an adjusted *p*-value < 0.05. DE genes were clustered by hierarchical clustering of log$_2$ transformed expression and displayed using MeV v.4.9[172]. GO enrichment analyses were done with BINGO[159,160] using a Benjamini and Hochberg false discovery rate correction[161] of 0.05.

Overrepresentation of MapMan[33,34] bins were tested with Wilcoxon Rank Sum test and corrected for multiple testing with the Benjamini Hochberg correction with a *p*-value < 0.05.

## Plant membrane extraction, fractionation, and protein quantitation

Membrane preparations were obtained for *S. bigelovii* shoots treated with 0, 50, 200, or 600 mM NaCl for 6 weeks with three biological replicates per treatment and processed similarly as described in Dunkley, et al.[57] with all steps done at 4 °C. 100–120 g of shoot tissue were blended in homogenization buffer (5% glycerol, 100 mM KCl, 0.5% PVP, 10 mM EDTA, 2 mM PMSF, 10% sucrose, 50 mM HEPES, pH 7.8) and osmotically adjusted with sucrose to shoot sap osmolality measured with a VAPRO® vapor pressure osmometer (ELITechGroup Biomedical Systems). The homogenate was filtered through three layers of cheesecloth and spun at 10,000 x *g* for 30 min to sediment debris and intact organelles. The supernatant containing the

microsomal fraction was transferred to a new tube and spun at 100,000 x *g* for 1 h. The pellet was resuspended in 2 mL resuspension buffer (5% glycerol, 50 mM KCl, 10% sucrose, 25 mM HEPES, pH 7.8) and layered on top of a 24 mL continuous sucrose density gradient (20 – 45%). Gradients were spun at 100,000 x *g* for 14 h. Fractions were recovered each 2 mL with an Auto Densi-Flow II (Buchler Instruments) resulting in 12 different density fractions. Fraction density was measured with a digital refractometer (Brix/RI-Chek, Reichert). To release cytosolic proteins trapped inside microsomes, fractions were diluted 3 times in 160 mM $Na_2CO_3$, left on ice for 30 min, and spun at 100,000 x *g* for 1 h. The pellet was washed with 160 mM $Na_2CO_3$ and spun again at 100,000 x *g* for 1 h. The pellet was washed with water and spun at 100,000 x *g* for 1 h. The resultant pellet was resuspended in 1 mL of storage buffer (7 M urea, 2 M thiourea, 5 mM Tris-HCl, pH 7.1) and stored at −80 °C. Protein concentrations were measured by the Bradford assay[173].

## Protein digestion and LC-MS analyses

Proteins for each fraction were trypsin digested based on filter aided sample preparation[59]. 30 μg of protein were incubated with 100 mM DTT for 1 h at 37 °C. Samples were loaded into a Vivacon 500 hydrosart 30 kDa filter (Sartorius) and spun at 14,000 x *g* for 15 min. 100 μL of 50 mM iodoacetamide were added to the column, incubated for 20 min, and spun at 14,000 x *g* for 10 min. 100 μL of 8 M urea, 100 mM Tris-HCl, pH 7.8 were added and the sample was spun at 14,000 x *g* for 15 min. This step was repeated twice. 40 μL of trypsin 1:100 (Sequencing grade modified trypsin, Promega) in 50 mM ammonium bicarbonate were added to the sample and incubated in a wet chamber at 37 °C overnight. Samples were spun at 14,000 x *g* for 10 min. 100 μL of 50 mM ammonium bicarbonate were added to the column and spun at 14,000 x *g* for 15 min. 50 μL of 0.5 M NaCl were added to the column and spun at 14,000 x *g* for 15 min followed by the addition of 50 μL ammonium bicarbonate and another centrifugation step at 14,000 x *g* for 15 min. Filtered peptides were acidified by the addition of 2% trifluoroacetic acid (TFA) and desalted with Sep-Pak C18 cartridges (Waters Scientific). Samples were lyophilized, resuspended in 15 μL 3% acetonitrile 0.1% TFA, and quantified with Nanodrop measurement at 280 nm wavelength. 1 μg of peptides for label-free LC/MS analyses were loaded to an Ultimate 3000 UHPLC/Q-Exactive system (Thermo Scientific) with an Acclaim PepMap RSLC 75 μm × 15 cm nanoViper column and run for 1 h.

## MS spectra analyses and protein abundance quantitation

Mass spectra peak files were compared against a combined set of *S. bigelovii* and *S. europaea* proteins derived from their genome annotation with Mascot v. 2.8.0.1[174]. Database searching parameters included trypsin as digestion enzyme, peptide tolerance of ± 10 ppm, fragment tolerance of ± 0.5 Da, carbamidomethyl of cysteine as fixed modification, oxidation of methionine as variable modification, two maximum missed cleavage sites, and reverse sequences as decoy. Mascot files were loaded into Scaffold v.4.8.8[175] for protein quantitation and differential abundance analyses. For differential abundance analyses, Mascot files were loaded as Multidimensional Protein Identification Technology (MudPIT) with three biological replicates per treatment. Protein identification required a minimum of two assigned peptides per protein, a greater than 95% peptide probability by the Scaffold Local FDR algorithm, and a greater than 99% probability of protein identification (Source Data file). Protein abundances were normalized by total spectrum counts. Differential protein abundances were tested at a significance of $p \le 0.05$ and corrected for multiple testing with the Hochberg-Benjamini correction[161]. Differentially abundant proteins were analyzed for pathway and overrepresentation with MapMan[33,34,176] and GO enrichment analysis was performed with BINGO[159].

## Subcellular localization prediction

Protein subcellular localization was predicted with the pRoloc[65–70] R package. To create a *Salicornia* organelle marker set, protein sequences of *S. bigelovii* and *S. europaea* genomes were combined. The *Salicornia* protein dataset was blasted against 279 Arabidopsis organelle markers retrieved from pRolocdata[57,60–64]. Positive hits with Arabidopsis markers were in silico predicted for subcellular compartmentalization with mGOASVM Plant V2[177]. Proteins whose predicted subcellular localization by mGOASVM Plant V2 was congruent with their Arabidopsis organelle marker blast hit, were assigned as *Salicornia* organelle markers. To predict for protein subcellular localization based on protein abundance profiles in the 12 density fractions, protein abundances per fraction of samples coming from shoots of *S. bigelovii* plants treated with 0, 50, 200, and 600 mM NaCl were analyzed with pRoloc. pRoloc was run under a supervised maximum likelihood (ML) analysis with a support vector machines analysis with 1000 iterations and using the generated *Salicornia* organelle marker set as reference.

## Validation of subcellular compartmentalization prediction of pRoloc by Western blot

The predicted subcellular localization by pRoloc was validated with western blots against organelle markers. 5 μg of protein per density fraction were separated in a 4–20% SDS-PAGE gel and used for western blot analyses using antibodies for different organelles (polyclonal antibodies generated in rabbit, Agrisera): Chloroplast PsbA (catalog number AS05 084) 1:10000 working dilution; Plasma membrane $H^+$ ATPase (catalog number AS07 260) 1:1000 working dilution; and Tonoplast V-ATPase (catalog number AS07 213) 1:2000 working dilution. Signal was detected by chemiluminescence using WesternBreeze® Chemiluminescent Kit anti-rabbit (Invitrogen) and according to the manufacturer's instructions. Western blot profiles for different subcellular localization marker proteins were compared against protein profiles generated with pRoloc.

## Subcellular localization assays in tobacco leaves

To assess the subcellular localization of SbiSOS1, reporter fusions were generated and transiently expressed in tobacco leaves. To generate the constructs, the CDS of SbiSOS1 and AtSOS1 (At2g01980) were synthesized by GenScript and cloned into the pGGC000[178] entry vector and the modules were cloned into the pGGZ001 destination vector (pGGC000 and PGGZ001 were a gift from Jan Lohmann, Addgene plasmid #48858; http://n2t.net/addgene:48858; RRID:Addgene_48858 and Addgene plasmid #48868; http://n2t.net/addgene:48868; RRID:Addgene_48868). The Arabidopsis plasma membrane intrinsic protein 1;4 (AtPIP1;4) and the vesicle associated membrane protein (AtVAMP711)[76] were used as plasma membrane and tonoplast markers, respectively. The overexpression constructs *35 S::SbiSOS1-eGFP*, *35 S::eGFP-SbiSOS1*, *35 S::AtSOS1-eGFP*, *35 S::mCherry-AtPIP1;4*, *35 S::mCherry-AtVAMP711*, and *35 S::eGFP* (as a negative control) were assembled by GreenGate cloning[178]. All constructs were transformed into the *Agrobacterium tumefaciens* strain GV3101[179] carrying the pSOUP plasmid[180] by electroporation[181]. Leaves from 4-week old *Nicotiana benthamiana* plants were inoculated with *A. tumefaciens* culture mixtures as follows: protein of interest (SbiSOS1 or eGFP), marker protein (AtPIP1;4 or AtVAMP711), and p19 (viral gene silencing suppressor to enhance transient expression efficiency)[182]. After 96 h of agroinfiltration, the subcellular localization of SbiSOS1 and AtSOS1 were determined by their co-localization with the two markers in leaf epidermal cells treated with 0.3 M D-mannitol for 3 h by confocal microscopy. Images were captured using a ZEISS LMS 710 microscope and analyzed by ZEN 2.3 SP1 and Fiji[183]. The excitation/emission wavelengths used in the confocal microscopy were 594/580-630 nm for mCherry and 488/500-530 nm for eGFP.

### RNA extraction and cloning of candidate genes into *E. coli*

RNA was extracted from shoots of *S. bigelovii* plants grown in 200 mM NaCl using TRIzol™ with an RNA MiniPrep kit (Zymo Research, USA), following the manufacturer's protocol. cDNA was prepared from the extracted, good quality RNA with SuperScript™ II (Invitrogen, USA), following the manufacturer's protocol. Genes were amplified by PCR using the high fidelity KOD DNA polymerase (Toyobo, Japan), cloned into pCR™8/GW/TOPO® (Invitrogen, USA), and transformed into *E. coli*. Since *S. bigelovii* is an autotetraploid undomesticated species, several isoforms per gene might be cloned. Entry clones were sequenced and compared with the reference transcriptome. Genes that did not match the sequence were re-amplified and cloned again. If the same polymorphisms were observed in subsequent amplifications, they were considered natural allelic variants and proceeded to further analysis. In the case two or more alleles were observed, alleles sharing equal sequences with *S. brachiata* or *S. europaea* were selected as it was assumed that conserved alleles across species would be more likely to be functional. Entry clones with and without stop codons per gene were recombined with Gateway™ LR Clonase™ II (Invitrogen, USA) into pAG426GAL-ccdB-EGFP. pAG426GAL-ccdB-EGFP was a gift from Susan Lindquist (Addgene plasmid # 14203).

### Yeast transformation, spot assays, and subcellular localization

Plasmids were transformed into the salt sensitive AXT3 (Δena1-4::HIS3 Δnha1::LEU2 Δnhx1::TRP1) yeast strain by the lithium acetate method[184]. Yeast spot assays were carried out in synthetic defined (SD) medium: 1.7 g/L BD™ Difco™ yeast nitrogen base without amino acids and ammonium sulfate, 5 g/L ammonium sulfate, 0.77 g/L complete supplement mix (CSM) Drop-out – uracil (Formedium), adjusted to pH 5.6 with KOH, and supplemented with either 2% glucose or galactose; or in arginine phosphate (AP) medium[185]: 10 mM L-arginine, 8 mM phosphoric acid, 2 mM MgSO$_4$, 0.2 mM CaCl$_2$, 20 μg/L biotin, 2 mg/L calcium pantothenate, 2 μg/L folic acid, 400 μg/L niacin, 200 μg/L riboflavin, 10 mg/L inositol, 200 μg/L p-aminobenzoic acid, 400 μg/L ZnSO$_4$, 400 μg/L pyridoxine hydrochloride, 400 μg/L thiamine hydrochloride, 40 μg/L CuSO$_4$, 500 μg/L H$_3$BO$_3$, 100 μg/L KI, 200 μg/L FeCl$_3$, 200 μg/L Na$_2$MoO$_4$, 1 mM KCl, 0.77 g/L complete supplement mix (CSM) Drop-out – uracil (Formedium), adjusted to pH 6.5 with L-arginine, and supplemented with either 2% glucose or galactose.

Yeast spot assays were done with three different colonies per construct. In short, to normalize growth, yeast transformed with pAG426GAL-gene-EGFP were grown in SD or AP medium with 2% glucose for 2 days at 30 °C. 200 μL of the pre-grown yeast were transferred into 1.8 mL of medium with 2% glucose and grown for 16 h at 30 °C. The OD$_{600}$ was measured and each sample was adjusted to an OD$_{600}$ of 0.6 by diluting with medium without any carbon source. For each sample, four 10-fold dilutions were prepared. 10 μL per sample were spotted in each SD or AP plate. SD plates consisted in SD medium with 2% agar and supplemented with either 0, 100 mM, 150 mM, 175 mM, and 200 mM NaCl or 800 mM KCl. SD plates were supplemented with 2% galactose for all treatments and with 2% glucose for plates with 0 mM NaCl as control. AP plates consisted of AP medium with 2% agar and supplemented with either 0, 20 mM, 30 mM, 40 mM, and 60 mM NaCl. AP plates were supplemented with 2% galactose for all treatments and with 2% glucose for plates with 0 mM NaCl as control. Plates were grown at 30 °C for 2–5 days. Subcellular localization of proteins was analyzed by EGFP fusions in the C′ terminus of each protein and observed with a confocal laser scanning microscope (LSM 710, Zeiss, Germany). Images were acquired with Zen 2009 image software (Zess, Germany) and analyzed with Fiji image analysis software[183].

To assess the subcellular localization of SbiSALTY *in planta*, *SbiSALTY* was cloned into the pBWA(V)HS-GLosgfp vector and transfected into rice protoplasts as described in Zhang, et al.[186]. The ER-mCherry marker ER-rk (TAIR: CD3-959)[187], which consists of the AtWAK2 signal peptide at the N-terminus fused to the reporter protein mCherry with the ER retention signal His-Asp-Glu-Leu at its C-terminus, and empty pBWA(V)HS-GLosgfp vector were used to confirm the subcellular localization of SbiSALTY.

### Protein expression and purification

cDNA encoding SALTY was codon optimized (Integrated DNA Technologies, Inc.), subcloned into the Expresso® SUMO Cloning system, a modified PET32a vector with an N-terminal His$_6$-SUMO expression tag and SUMO protease cleavage site (Lucigen Corporation). SALTY fusion was expressed in Codon + BL21 (DE3) *E. coli* cells from Merck (Cat# CMC0014). Cells were grown in Luria broth or labeled M9 medium with kanamycin (His$_6$-SUMO-SALTY). The pre-culture and unlabeled cell growth were accomplished using the auto-induction method[188]. The cell culture media were grown and suspended until OD = 0.6, then induced with 0.5 mM IPTG and incubated at 25 °C for 24 h. Cells were collected and resuspended in 50 mL lysis buffer (50 mM Tris-HCl, pH 7.5, 250 mM NaCl, 30 mM Imidazole, 1 mM TCEP for His$_6$-tag recombinant proteins) per liter of culture. 0.5 mM phenylmethylsulfonyl fluoride (PMSF) and 1 mM β-mercaptoethanol (β-ME) were added to the lysis buffer. Cells were lysed using a cell disruptor (Constant Systems) and DNase I and RNase A (Invitrogen, Cat# 12091039 and AM2224) were added to digest the DNA/RNA. After centrifugation, the clarified lysate was applied to a HisTrap HP (GE Healthcare) and eluted with elution buffer (Lysis Buffer + 0.5 M Imidazole). To remove the His$_6$-SUMO tag, the protein was cleaved with SUMO protease (Invitrogen, Cat# 12588018), and purified using an AKTA with a HisTrap HP column. The uniform $^{15}$N-labeling of recombinant protein was done according to well-established protocols[189].

### Sequence analysis and circular dichroism spectroscopy

In silico prediction for intrinsically disorder domains of SbiSALTY protein was done with DisEMBL v.1.5[190]. Coiled-coils region prediction was done with COILS[191]. The circular dichroism (CD) spectra of SALTY (10 μM in 50 mM MES, 50 mM NaCl, pH 6.5) were acquired using a Jasco 1500 spectropolarimeter flushed with N$_2$ and a cuvette with 0.1 cm path length. The spectra to investigate its secondary and tertiary structure were measured from 190 – 260 nm and 250 – 350 nm with 0.2 nm intervals. The temperature stability was assessed with temperatures ranging from 25 – 90 °C at three wavelengths, 201, 208 and 222 nm. The spectrum of the buffer was subtracted. The internal software and CAPITO (CD Analysis and Plotting Tool) were used to assess the secondary structure elements of fused SALTY based on its CD spectrum[192].

### Protein NMR experiments

The NMR samples of $^{15}$N-SALTY$_{1-357}$ contained 50–1000 μM protein in NMR buffer (50 mM MES, 50 mM NaCl, pH 6.5, 1 mM TCEP and 5%/95% D$_2$O/H$_2$O *v/v*). All NMR spectra were acquired at 5°C on an 800 MHz Bruker Advance NEO spectrometer equipped with TCI $^1$H&$^{19}$F/$^{13}$C/$^{15}$N cryogenic probe head, running Topspin version 4.0.7.

### Reporting summary

Further information on research design is available in the Nature Portfolio Reporting Summary linked to this article.

## Data availability

All sequencing data are available in the NCBI Short Read Archive (SRA), Bioprojects PRJNA733891, and PRJNA733892. Genome assemblies and annotation data can be found at www.salicorniadb.org. Protein data are available at ProteomeXchange identifier PXD051103. Source data are provided with this paper.

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

## Acknowledgements

Research reported in this publication was supported by the King Abdullah University of Science and Technology (KAUST). We thank E. Glenn at University of Arizona and M. Sagi at Blaustein Institute for Desert Research for their assistance in providing seeds. We also thank E. Martinoia at University of Zurich for his technical advice on gradient formation for membrane separation. We would like to thank our visiting student, J. Otero, for his assistance in the laboratory.

## Author contributions

M.T., N.V.F., O.R.S., and S.M.S. conceived the project and experiments. M.T. and S.M.S. supervised the research. O.R.S. and V.J.M. generated the sequencing data. D.B., E.H., and J.C. performed the karyotyping and flow cytometry. O.R.S. performed the bioinformatic analyses and genome assemblies. M.A. and O.R.S. evaluated the genome assemblies. O.R.S. performed the phenotypic, transcriptomic, and proteomic experiments and analyses. M.P.R., O.R.S., and V.J.M overlooked the plant material. J.P.A.V. and V.J.M planned the tobacco subcellular localization assays. J.P.A.V. and L.M.C.L. performed the tobacco subcellular localization assays. O.R.S. performed the salt tolerance assays in yeast. L.J., and M.J. led the protein structural analyses. K.C. and O.R.S. did the protein production and purification. K.C., L.J., and M.J. performed the NMR and CDD analyses. K.C. performed the rice protoplast transformation. All authors contributed to the writing of the manuscript. M.T., O.R.S., and S.M.S. organized the manuscript.

## Competing interests

The authors declare no competing interests
