## [Peer Review File · Nature Communications]

SOS1 tonoplast neo-localization and the RGG protein SALTY are important in the extreme salinity tolerance of *Salicornia bigelovii*Reviewer #1 (Remarks to the Author):

The manuscript by Salazar et al., provides a genomic, transcriptomic, and proteomic resource to study salt adaptive mechanisms in halophyte species: *Salicornia bigelovii* and *Salicornia europaea*. The main finding of the transcriptomic studies is that *Salicornia* appears to be able to maintain a growth-associated transcriptional program even at the highest salt concentrations, while low concentrations may induce a nutrient starvation-like response, perhaps due to the requirement for salinity for normal ion balance. Through their organelle proteomics analysis, they were able to characterize the proteins associated with different cellular compartments and characterize changes induced by salinity. These data allowed for the identification of a novel protein named SALTY and characterized as an intrinsically disordered protein via NMR. SbiSALTY seems to localize to the endoplasmic reticulum under 600 mM NaCl treatment and increase salt tolerance in yeast grown in media supplemented with up to 250 mM NaCl. This coincides with previous work by Ambrosone et al., showing that *Arabidopsis* RGG overexpression lines also lead to increased salt tolerance by improving water retention mediated by ABA-dependent stomatal closure.

An unresolved aspect of the work presented in this manuscript relates to the role of the *Salicornia* ortholog of SOS1 (SbiSOS1). In *Arabidopsis* and related halophyte species SOS1 orthologues are localized to the plasma membrane and function to export sodium ions out of the cytoplasm. In Salazar et al., proteomics experiments suggest that SbiSOS1 is localized to the tonoplast membrane. Yeast assays suggest SbiSOS1 is lethal to *E. coli*. The authors hypothesize that the SbiSOS1 is localized to the tonoplast and may facilitate accumulation and sequestration of sodium ions to the vacuole. This may be indicative of diverging salt adaptive mechanisms employed by the *Amaranthaceae* family and would be an extremely exciting result, however, a few key experiments are missing here that could substantially strengthen the conclusions that can be made here (see specific points below).

Overall, their genomics and transcriptomics research reveal insights into the incredible salt tolerance displayed by *Salicornia* species. I have a few concerns about the clarity of the text and some suggestions for follow-up experiments/text that could strengthen the paper.

Comments/Questions:

- 1) The authors show that SbiSOS1 was one of the most abundant proteins in shoots and exhibits tonoplast association. These data support a very exciting model that SbiSOS1 may protect *Salicornia* from salinity by promoting the accumulation of sodium in the vacuolar compartment, however, this model is largely untested. It would be relatively easy to test the intrinsic ability of SbiSOS1 to localize to the tonoplast by making an *Arabidopsis* line expressing a fluorescently tagged version of the protein or through transient expression (tobacco leaf or protoplasts). This experiment would allow the authors to confirm the change in transporter localization and test whether this property is intrinsic to SbiSOS1 coding sequence rather than a change in protein trafficking that is mediated by other factors. Further characterization of the salt tolerance of these lines would be nice, but not necessary for the scope of the current work. Finally, it is not clear if there are any sequence features that explain the new localization pattern of SbiSOS1 and this should be analyzed in the current work.
- 2) There is little comparison between the transcriptomic and proteomic data. While it is appreciated that these two methods reveal different regulatory mechanisms, a discussion of the similarities and differences would still be helpful.
- 3) Figure 5 shows yeast spot assays for genes predicted to play a role in salt tolerance in *Salicornia* but these assays don't say much about sodium accumulation in yeast. Adding quantitative data of ion levels like in Fig.2, Shi et al. (2002), may reveal more about how this transgene is acting to increase stress tolerance. There is also no data shown for SbiNHX4 and SbiKAT2 at higher salt concentrations + Galactose treatments, even though these seem to also increase salt tolerance in yeast to similar levels as SbiSALTY in Panel a.
- 4) Figure 6 shows expression of SbiSALTY in rice protoplasts and the authors conclude that it localizes to the endoplasmic reticulum. However, the PCA plot in Supplementary Figure 11 shows that SbiSALTY groups closer to thylakoid/mitochondrion clusters. What is the explanation for this? Furthermore, what is the justification for using rice (monocot) protoplasts instead of another closely related eudicot? The figure legend should state exactly which protein sequence was used

for the ER-localized control. Finally, the micrographs should use a color-blind compatible color scheme to show the fluorescence data.

5) Since the authors show that SbiSALTY protein abundance is strongest under 600 mM NaCl treatments, does it follow that it only plays a role at higher concentrations of NaCl in *Salicornia*? More speculation as to how SbiSALTY helps with salt tolerance and the role of RGG proteins would be helpful.

6) Did the authors look at other genes encoding non-transporters besides RGG's? Since GO enrichment analysis revealed processes related to high light intensity, toxin metabolism, etc.?

Minor:

7) Title: While catchy, seems more appropriate for a review article. I suggest a title that highlights the most important keywords of the study.

8) Abstract: Consider rephrasing sentence 3 so that "much can be learnt from them" is at the beginning of the sentence.

9) Introduction: Rephrase sentence 2

10) Results; NaCl induces expression of genes associated with growth; "We then looked at...gene expression profiles" should be addressing (Fig.3d-f) instead of (Fig.3b-d).

11) Results; The RGG protein SALTY is an intrinsically disordered protein and is localized to the ER; figure references switch from 6 to 2. I believe all the figure references should be to figure 6.

12) Page 9, second paragraph, first sentence needs a reference.

13) Supplementary figure 1b: the y-axis label provides units but the variable is unitless.

14) Supplementary figure 12 is missing panel letters and both panels are confusing as they lack y-axis labels.

15) Supplementary figure 7b: are the images used copyright protected? These look like images from a textbook.

16) Discussion: No citation for "In *Arabidopsis*, the overexpression of AtRGGGA, encoding an RGG protein, increased salt and drought tolerance compared to wild type plants." (Ambrosone et al., 2015).

Reviewer #2 (Remarks to the Author):

In this study, the authors analyzed the phenotypic responses and Na⁺/Cl⁻/K⁺ content of *S. bigelovii* to different concentrations NaCl. Combined the first genomic resources generated by the authors with other four plant families for Gene orthology analysis. Subsequently, they conducted integrative analyses of RNA-seq and organellar membrane proteome under normal and salt conditions for salt tolerance related genes and then identified in yeast. Generally, the study utilizes an interesting approach to identify and characterize genes involved in the salt resistance. However, there are several significant issues with this manuscript that need to be addressed.

The data presentation and manuscript readability should be improved before submission. I recommend that a thoroughly revised manuscript be resubmitted.

1. The phenotypic in Fig 1 without 5-week-old plants before NaCl treated for ensure consistent state of the plants, phenotypic data in Fig 1 should be updated. Both the result and method did not mention whether the phenotypic are repeatable.

2. The authors conclude that 200 mM NaCl as the optimum concentration for *S. bigelovii* growth, however, the shoot cross-sectional area and water content in shoots and roots at 600 mM NaCl was bigger or higher than 200 mM NaCl, maybe the optimum concentration is between 200 to 600, not the 200 mM. By the way, the optimum concentration standard is the plant height or plant biomass, why?

3. The number of differentially expressed genes between treatments at 200 mM and 600 mM with 6 weeks is only 8 genes, these genes can explain the difference of shoot cross-sectional area and ion concentrations?

4. Neo-localization of SOS1 to the tonoplast determine the gain function of it. However, the result need more sufficiently strong evidence to provide. The authors should express SOS1 in rice (or other) protoplasts suggests its localization and co-localized with a tonoplast marker, indeed, SOS1 could transport Na⁺ into tonoplast should verify by transport activity assay.

5. There is need more evidence to verify that DUF26 protein increased survival of plants exposed to severe NaCl treatments in *S. bigelovii*, also, SbiSALTY interact with ribosomes, the PCA analysis (Supplementary Fig.11) just a prediction assay.
6. To identify genes of *S. bigelovii* involved in salt tolerance, genes were selected for salt tolerance assays just in yeast, the authors should examine the salt tolerance functionality of several genes in genetically transgenic materials with overexpression or CRISPR in *S. bigelovii* or *Arabidopsis thaliana* or rice.
7. The authors need to verify that the figure legends provide all necessary details for the reader to understand the figures. A few examples- the ER marker in Figure 6g should detail which ER genes with which reference? Subcellular localization in yeast of Supplementary Figure 15/16/17/18 needs details and the picture arrangement should display in the same format (some 3 pictures and some 4)? Figure ordinate in Supplementary Figure 15c should align?
8. Citation of references, some titles may only have the first letter of the word capitalized, such as ref 6, 7, 9, while others may all have the first letter capitalized, such as ref13, 18, 20; some journal is abbreviated as ref7, 14, while others are spelled as ref12, 13.

Reviewer #3 (Remarks to the Author):

The manuscript examines *Salicornia* to understand its underlying ability to withstand high NaCl concentrations mainly by characterizing its transcriptome, followed by examining the effect of several proteins involved in salt response in a yeast model. Overall there is a considerable amount of work presented that is worthy of publication but I would not consider the work significant in that it feels preliminary with unfortunately only some insight into the molecular and cellular mechanisms.

One of the issues is the switching between several membrane associated proteins and one of the characterized soluble proteins by NMR. I understand that it is difficult to work with membrane proteins by NMR, but the emphasis in the early part of the results on membrane proteins and ion channels but then later switching to a soluble protein where the mechanism is unclear confused me. Also, the inclusion of NMR data and the observation that the protein is disordered did not show any significant mechanistic insight. The authors mention late embryogenesis proteins in the discussion, which are proteins known to be involved in stress tolerance, but do not appear to connect that to any of the identified proteins.

The second issue is that the comparison of the genes that increased salt tolerance versus those that decreased it. Again, a considerable amount of work has been performed but I have a difficult time connecting it to a mechanism. In Figure 5, there are genes that increased tolerance and those that decreased it. My interpretation is that the expression of some of these genes may be up-regulated and others are down-regulated, but I did not see an explanation of this in the results. The authors also point out in the discussion that the lack of protection in yeast may be due to different localization compared to plants. These results could be strongly support by several selective knockout studies in *Salicornia* to show that these proteins are actually important for salt tolerance.

Response to the Reviews

We would like to thank the reviewers for their time and effort put into reviewing our manuscript. We understand this is a time-consuming task and appreciate their input, as it helps us improve the manuscript from a scientific standpoint and makes it clearer for the readers.

Reviewer #1

The manuscript by Salazar et al., provides a genomic, transcriptomic, and proteomic resource to study salt adaptive mechanisms in halophyte species: *Salicornia bigelovii* and *Salicornia europaea*. The main finding of the transcriptomic studies is that *Salicornia* appears to be able to maintain a growth-associated transcriptional program even at the highest salt concentrations, while low concentrations may induce a nutrient starvation-like response, perhaps due to the requirement for salinity for normal ion balance. Through their organelle proteomics analysis, they were able to characterize the proteins associated with different cellular compartments and characterize changes induced by salinity. These data allowed for the identification of a novel protein named SALTY and characterized as an intrinsically disordered protein via NMR. SbiSALTY seems to localize to the endoplasmic reticulum under 600 mM NaCl treatment and increase salt tolerance in yeast grown in media supplemented with up to 250 mM NaCl. This coincides with previous work by Ambrosone et al., showing that *Arabidopsis* RGG overexpression lines also lead to increased salt tolerance by improving water retention mediated by ABA-dependent stomatal closure.

An unresolved aspect of the work presented in this manuscript relates to the role of the *Salicornia* ortholog of SOS1 (SbiSOS1). In *Arabidopsis* and related halophyte species SOS1 orthologues are localized to the plasma membrane and function to export sodium ions out of the cytoplasm. In Salazar et al., proteomics experiments suggest that SbiSOS1 is localized to the tonoplast membrane. Yeast assays suggest SbiSOS1 is lethal to *E. coli*. The authors hypothesize that the SbiSOS1 is localized to the tonoplast and may facilitate accumulation and sequestration of sodium ions to the vacuole. This may be indicative of diverging salt adaptive mechanisms employed by the Amaranthaceae family and would be an extremely exciting result, however, a few key experiments are missing here that could substantially strengthen the conclusions that can be made here (see specific points below).

Overall, their genomics and transcriptomics research reveal insights into the incredible salt tolerance displayed by *Salicornia* species. I have a few concerns about the clarity of the text and some suggestions for follow-up experiments/text that could strengthen the paper.

We appreciate the time and effort placed on this review, and we have tried to address the comments to the best of our ability. We address the specific comments below:

1) The authors show that SbiSOS1 was one of the most abundant proteins in shoots and exhibits tonoplast association. These data support a very exciting model that SbioSOS1 may protect *Salicornia* from salinity by promoting the accumulation of sodium in the vacuolar compartment, however, this model is largely untested. It would be relatively easy to test the intrinsic ability of SbiSOS1 to localize to the tonoplast by making an Arabidopsis line expressing a fluorescently tagged version of the protein or through transient expression (tobacco leaf or protoplasts). This experiment would allow the authors to confirm the change in transporter localization and test whether this property is intrinsic to SbiSOS1 coding sequence rather than a change in protein trafficking that is mediated by other factors. Further characterization of the salt tolerance of these lines would be nice, but not necessary for the scope of the current work. Finally, it is not clear if there are any sequence features that explain the new localization pattern of SbiSOS1 and this should be analyzed in the current work.

We agree with the reviewer's comment that a subcellular localization assay would strengthen the current work. To address this, we used transient expression assays in tobacco leaves, expressing SbiSOS1 GFP fusions at the N and C termini and assessing their colocalization with tonoplast and plasma membrane markers. In agreement with our hypothesis, we observed that SbiSOS1 localizes to the tonoplast in tobacco leaves, supporting our proteomics results (Fig. 5, Supplementary Figs. 12-14). We also observed its localization to the plasma membrane - this may reflect an *in planta* localization to both membranes, or may be due to spillover of protein to the plasma membrane due to high abundance of protein caused by the use of a 35s promoter, as the default pathway for membrane protein subcellular localization is the plasma membrane through the secretory pathway. We have included these comments in the discussion (Page 33 lines 19-23 to page 34 lines 1-5).

We have also included an analysis of SbiSOS1 sequence, where we aligned SOS1 sequences from several species and calculated the nonsynonymous (K_a) and synonymous (K_s) substitution rates across the *S. bigelovii* vs Arabidopsis SOS1 in sliding windows. We identified greater divergence at the termini of the protein, with one region near the C terminus of SbiSOS1 in particular showing a high K_a/K_s . These divergent regions may be involved in the vacuolar localization of SbiSOS1. We have included this analysis in the main text and in the results section (Page 22 lines 9-17) and discussion (Page 34 lines 6-10). Additionally, two Supplementary Figures (15 and 16) for this analysis were added.

2) There is little comparison between the transcriptomic and proteomic data. While it is appreciated that these two methods reveal different regulatory mechanisms, a discussion of the similarities and differences would still be helpful.

We have included a discussion about the differences and similarities of the transcriptomic and proteomic enrichment analyses under the subheading "NaCl increases the abundance of membrane proteins related to ATP synthesis" (Pages 24 and 25). While a direct comparison between both approaches might prove difficult, as the transcriptomics analysis uses the whole gene set available for *Salicornia* and the

proteomics data is derived from membrane protein enrichments (leaving out most soluble proteins), the overall responses were similar. In brief, increased salt concentrations promoted growth, photosynthesis, and ATP synthesis, while a lack of NaCl caused stress responses. However, some responses were only significant in the gene expression analysis.

After examining the data further, we realized that most of these terms were also present in the proteomics data prior to multiple testing correction, but were no longer significant after a Benjamini and Hochberg false discovery rate correction. We believe that the nature of working with membrane proteins, leading to a limited number of identified proteins, might result in reduced statistical power when compared to the transcriptomics analysis. However, it remains possible that these differences could reflect differences in responses of soluble vs membrane proteins to stress. Another factor to consider is the different turnover rate of mRNAs and their different translation rates, as RNAseq data and protein abundances are often poorly correlated.

3) Figure 5 shows yeast spot assays for genes predicted to play a role in salt tolerance in *Salicornia* but these assays don't say much about sodium accumulation in yeast. Adding quantitative data of ion levels like in Fig.2, Shi et al. (2002), may reveal more about how this transgene is acting to increase stress tolerance. There is also no data shown for SbiNHX4 and SbiKAT2 at higher salt concentrations + Galactose treatments, even though these seem to also increase salt tolerance in yeast to similar levels as SbiSALTY in Panel a.

We greatly appreciate the suggestion to include the ion accumulation in the yeast assays. As we decided to focus on SbiSALTY, which is a soluble protein not possessing any transmembrane domain and due to the nature of RGG proteins primarily binding DNA and RNA, we did not follow the ion accumulation in yeast. We do think, however, that measuring the ion accumulation would be informative in the case of the SbiNHXs, which showed very different phenotypes in yeast. We would like to include such analyses in a further publication focusing on these transporters. Regarding the request to show the growth of SbiNHX4 and SbiKAT2 at higher salt concentrations, yeast spot assays for both genes can be found in Supplementary Figure 18, where SbiNHX4 could no longer rescue yeast at the highest salinity, in contrast to SbiSALTY and SbiKAT2. This assay was done using the AP medium, which offers a harsher environment for yeast than the SD medium, hence the differences in NaCl concentrations. In the Supplementary Figure 19, we show the growth of SbiSALTY and SbiKAT2 at high potassium concentrations and it can be seen that the expression of SbiKAT2 results in cell death, probably due to high levels of potassium import, leading to accumulation of toxic concentrations of K^+ . Expression of SbiSALTY does not result in excess mortality.

4) Figure 6 shows expression of SbiSALTY in rice protoplasts and the authors conclude that it localizes to the endoplasmic reticulum. However, the PCA plot in Supplementary Figure 11 shows that SbiSALTY groups closer to thylakoid/mitochondrion clusters. What is the explanation for this? Furthermore, what is the justification for using rice (monocot) protoplasts instead of another closely related eudicot? The figure legend should state

exactly which protein sequence was used for the ER-localized control. Finally, the micrographs should use a color-blind compatible color scheme to show the fluorescence data.

The results of SbiSALTY in the PCA are to be taken with caution. While some traces of soluble proteins can be observed in our proteomics experiment, the experiment was designed to enrich for membrane proteins, greatly diminishing the abundance of soluble proteins. Since SbiSALTY is a soluble protein, its abundance is greatly reduced, making it difficult to reliably allocate it to a subcellular compartment. Since we hypothesize that SbiSALTY might bind to ribosomes, we included ribosomal proteins in the PCA plot in Supplementary Figure 11. It can be seen that ribosomal proteins also seem to be more abundant near the thylakoid/mitochondrion clusters and are found in the vicinity of SbiSALTY. Unfortunately, we did not add Mg^{2+} to our membrane fractions, which would have promoted the ribosomal assemblage to bind to the ER; hence, ribosomes appear scattered in our PCA, albeit around the theoretical density of the rough ER. For these reasons we believe that the subcellular localization assay in protoplasts is a more reliable assay to assess the subcellular localization of SbiSALTY than the PCA plot. These shortcomings are mentioned in the subheading “The RGG protein SALTY is an intrinsically disordered protein and is localized to the ER” (Page 30 lines 5-16), and in the discussion (Page 36 lines 12-15). We decided to use rice protoplasts to assess the subcellular localization of SbiSALTY as it was a system that we were familiarized with and was well established in our laboratory. As we considered that we were not working with a cell wall protein, we considered it to give us reliable information on its subcellular localization. We do agree that it would be best to observe its localization in *Salicornia* protoplasts; however, the system has not yet been fully developed. We appreciate the observation that the identity of the utilized ER marker was missing. The marker consists of the AtWAK2 signal peptide at the N-terminus followed by the mCherry protein fused with the ER retention signal at its C-terminus. We apologize for this omission; we have now included it in the figure legend and under methods. We thank the reviewer for pointing out the color-blind incompatibility of the fluorescence data. We have now included colorblind adjusted images for all localization images as Supplementary Figures (Figure 5 = Supplementary Figure 12; Supplementary Figure 13 = Supplementary Figure 14; and Figure 7 as Supplementary Figure 23).

5) Since the authors show that SbiSALTY protein abundance is strongest under 600 mM NaCl treatments, does it follow that it only plays a role at higher concentrations of NaCl in *Salicornia*? More speculation as to how SbiSALTY helps with salt tolerance and the role of RGG proteins would be helpful.

SbiSALTY gene expression and protein abundance seem to increase in response to increased NaCl. However, SbiSALTY is also expressed at lower NaCl concentrations, making it difficult to conclude if it is only being used when the cell is exposed to increased NaCl concentrations. Nevertheless, it raises the question of why this protein becomes more abundant upon salinity and is not always at higher abundances, as *Salicornia* grows in saline environments. RGG proteins have been found to be important in the formation of membrane-less compartments such as RNA-bodies. It was recently

found that three *Arabidopsis* RGG proteins bind to ribosomal proteins of the 40s and 60s subunits. SbiSALTY possesses a high degree of similarity with these three proteins, and it is likely that SbiSALTY also binds to ribosomes. It could well be that SbiSALTY possesses a “penalty” and this tradeoff becomes beneficial at higher NaCl concentrations. For instance, we propose that SbiSALTY binds to ribosomes and stabilizes them upon salt stress. Perhaps, the binding of SbiSALTY to the ribosomes partially decreases ribosomal performance at low NaCl concentrations but its protective effect allows ribosomes to function at higher efficiencies at higher salinities than without their interaction with SbiSALTY. We have included more speculation about the role of SbiSALTY in salt stress into the discussion (Page 35 lines 21-23, page 36 lines 1-2, 12-15 and 21-23, and page 37 lines 1-2 and 6-11).

6) Did the authors look at other genes encoding non-transporters besides RGG's? Since GO enrichment analysis revealed processes related to high light intensity, toxin metabolism, etc.?

SbiSALTY was included in the analysis due to its gene expression and protein abundance pattern, but our primary focus was to study transporter proteins. We could say that we were a bit lucky to identify SbiSALTY! Unfortunately, we didn't include other soluble proteins in our yeast assays, except for SbiIDP1, which is another intrinsically disordered protein, albeit lacking the RGG motif, which had a very negative effect in yeast. We do think that there is still a lot to be discovered regarding *Salicornia*'s extreme salt tolerance and proteins that are not transporters are likely to also play a key role. Future studies analyzing soluble proteins could provide us with hints into the key proteins involved in the response to stress in *Salicornia*. As the reviewer mentioned, *Salicornia* must not only have adaptations to increased salinity, but might have interesting mechanisms for tolerance to high light, heat, toxin metabolism, and others. For example, *Salicornia* has a high accumulation of saponins (not shown in this study), which are used for plant defense against pests and pathogens, but are also studied for their anti-inflammatory effects in humans. We think that our study serves as the basis for future studies in *Salicornia* and will hopefully pique the interest of researchers to further study this remarkable plant.

Minor:

7) Title: Title While catchy, seems more appropriate for a review article. I suggest a title that highlights the most important keywords of the study.

We have now changed the title into something more in line with the main findings of the study. The new title is: “Insights into the remarkable salinity tolerance of *Salicornia bigelovii*: roles for the RGG protein SALTY, and neo-localization of SOS1 to the tonoplast”.

8) Abstract: Consider rephrasing sentence 3 so that “much can be learnt from them” is at the beginning of the sentence.

We have modified the abstract and we hope that it better represents the study.

9) Introduction: Rephrase sentence 2

We have rephrased sentence 2.

10) Results; NaCl induces expression of genes associated with growth; “We then looked at...gene expression profiles” should be addressing (Fig.3d-f) instead of (Fig.3b-d).

We apologize for this mistake; we have fixed the text and it now points towards the correct figure panels (Page 14 line 21).

11) Results; The RGG protein SALTY is an intrinsically disordered protein and is localized to the ER; figure references switch from 6 to 2. I believe all the figure references should be to figure 6.

We have fixed the text and it now points to the correct figure, now Figure 7 due to the inclusion of the SbiSOS1 localization assay figure.

12) Page 9, second paragraph, first sentence needs a reference.

We have included a couple of references regarding the association of NHX proteins to salt tolerance (Page 10 line 12).

13) Supplementary figure 1b: the y-axis label provides units but the variable is unitless.

We have removed the units in Figure 1b, as it is a ratio.

14) Supplementary figure 12 is missing panel letters and both panels are confusing as they lack y-axis labels.

We have included headings and panel letters in Supplementary Figure 17 (previously Supplementary Figure 12), we hope the figure is more legible now.

15)Supplementary figure 7b: are the images used copyright protected? These look like images from a textbook.

We have changed the images for others generated by ourselves.

16) Discussion: No citation for “In Arabidopsis, the overexpression of AtRGGA, encoding an RGG protein, increased salt and drought tolerance compared to wild type plants.” (Ambrosone et al., 2015).

We had previously added the reference in the next sentence, but it is now being mentioned at the correct location (Page 36 line 1).

Reviewer #2

In this study, the authors analyzed the phenotypic responses and Na⁺/Cl⁻/K⁺ content of *S. bigelovii* to different concentrations NaCl. Combined the first genomic resources generated by the authors with other four plant families for Gene orthology analysis. Subsequently, they conducted integrative analyses of RNA-seq and organellar membrane proteome under normal and salt conditions for salt tolerance related genes and then identified in yeast. Generally, the study utilizes an interesting approach to identify and characterize genes involved in the salt resistance. However, there are several significant issues with this manuscript that need to be addressed. The data presentation and manuscript readability should be improved before submission. I recommend that a thoroughly revised manuscript be resubmitted.

We appreciate the time and effort placed on this review, and we have tried to address the comments to the best of our ability. We have revised the manuscript and have included changes that will hopefully improve the readability and quality of the manuscript. We address the specific comments below:

1. The phenotypic in Fig 1 without 5-week-old plants before NaCl treated for ensure consistent state of the plants, phenotypic data in Fig 1 should be updated. Both the result and method did not mention whether the phenotypic are repeatable.

We apologize for any possible confusion. *Salicornia* plants germinated for this experiment all came from the same seed stock derived from one inbred line, were germinated at the same time, and were grown under the same conditions. Three trays per treatment were grown for a total of 12 trays. We repeated the phenotypic assay under the same conditions twice to ensure consistent responses. Furthermore, we also observed the beneficial effect of NaCl in the growth of *Salicornia* across multiple other experiments. This is in agreement with several other studies where they found similar effects of NaCl to *Salicornia* growth and Na⁺ ion accumulation, such as Kong and Zheng, 2014, Lv et al., 2012, and Otori and Fujiyama, 2011. We have included additional information regarding the reproducibility in Figure 1 and under Methods (Plant material and growth conditions).

2. The authors conclude that 200 mM NaCl as the optimum concentration for *S. bigelovii* growth, however, the shoot cross-sectional area and water content in shoots and roots at 600 mM NaCl was bigger or higher than 200 mM NaCl, maybe the optimum concentration is between 200 to 600, not the 200 mM. By the way, the optimum concentration standard is the plant height or plant biomass, why?

The reviewer is right in pointing out that it is difficult to identify the exact optimum NaCl concentration with the available data. For a full lifecycle, ~200 mM NaCl seems to produce the highest amount of biomass (as means of dry mass). In this experiment we did not reach its full life cycle and due to the limited number of NaCl treatments the

exact optimum concentration cannot be established. We defined approximate optimum concentration for growth based on plant height and lateral branch formation, as *Salicornia* seeds are produced within its stem and branches. We have corrected the text to say: “suggesting that a concentration of NaCl somewhere between 200 mM – 600 mM NaCl is optimal for the growth of *S. bigelovii*, which is in agreement with previous studies” (Page 6 lines 1-3).

3. The number of differentially expressed genes between treatments at 200 mM and 600 mM with 6 weeks is only 8 genes, these genes can explain the difference of shoot cross-sectional area and ion concentrations?

We were also surprised with this result. However, both the transcriptomics and proteomics experiments were consistent in showing very few differences between plants treated with 200 mM and 600 mM NaCl for 6 weeks. This is in stark difference with plants treated with 200 mM and 600 mM NaCl for 1 week, where plants treated with 600 mM NaCl had the highest number of differentially expressed genes relative to any other treatment and seemed to suffer from a shock response. We believe that *Salicornia* plants became acclimated to the environment and synthesized the required proteins to handle the increased concentration of NaCl. Other proteins might be already expressed and be limited by the substrate availability rather than their abundances. For example, the water channels plasma membrane intrinsic proteins (PIPs) and tonoplast intrinsic proteins (TIPs), were amongst the most abundant membrane proteins in our study and may be important for the succulence of *Salicornia* - however, the abundance of these proteins remained fairly constant across treatments. Additionally, many transporters are regulated through post-translational modifications such as phosphorylation and might be crucial in the modulation of their activities. In this study, we didn't measure such modifications, as a different methodology would be required for the enrichment of phosphorylated proteins. It could very well be that differences in post-translational modifications contribute to NaCl responses. It would be interesting to assess post-translational modifications in *Salicornia* in future studies. We have now included some of these points in the discussion (Page 33 lines 1-7).

4. Neo-localization of SOS1 to the tonoplast determine the gain function of it. However, the result need more sufficiently strong evidence to provide. The authors should express SOS1 in rice (or other) protoplasts suggests its localization and co-localized with a tonoplast marker, indeed, SOS1 could transport Na⁺ into tonoplast should verify by transport activity assay.

We have now included a localization assay for SbiSOS1, where we transiently expressed the fusions *SbiSOS1::GFP* and *GFP::SbiSOS1* and compared their localization with a tonoplast and a plasma membrane marker in tobacco leaves. Congruently with the neo-localization hypothesis, we found that SbiSOS1 can indeed localize to the tonoplast in tobacco cells (Fig. 5, Supplementary Figs. 12-14). Further details are given in our response to Reviewer #1, Comment 1.

Regarding possible SbiSOS1 transport assays. Due to the apparent vacuolar localization of SbiSOS1, transport assays in heterologous systems would require a stable transformation of SbiSOS1 and multiple knock-outs in other vacuolar cation-H⁺ antiporters, such as those of the NHX family. This would require a substantial amount of time and we do not know the effects that SbiSOS1 could have in the development of the selected heterologous plant system. Alternatively, transport assays could be done in artificial liposomes; however, SOS1 needs to be activated through phosphorylation via SOS2/SOS3, which makes this experiment difficult. This could be overcome through hyperactive SOS1 mutants by the deletion of its C-terminal. However, we would prefer to measure its activity in its native form before assessing protein modifications. Ideally, SbiSOS1 activity should be tested in *Salicornia* mutants, unfortunately there are currently no methods available for *Salicornia* transformation. Nevertheless, we also agree with the importance of assessing SbiSOS1 transport activity and we hope to be able to show it in future studies.

5. There is need more evidence to verify that DUF26 protein increased survival of plants exposed to severe NaCl treatments in *S. bigelovii*, also, SbiSALTY interact with ribosomes, the PCA analysis (Supplementary Fig.11) just a prediction assay.

We agree that more evidence is required to confirm these hypotheses, we have just identified and suggested a few mechanisms that might help in the response to high concentrations of NaCl in *Salicornia*. Nevertheless, more studies are required to properly understand these responses. The enrichment in the downregulation of DUF26 proteins is an observation and further studies are required to confirm its involvement in salt stress. However, a previous study showed that the downregulation of *OsRMC* in rice (a DUF26 protein) lead to increased survival of rice upon salinity. These results suggest that it might be worth studying the involvement of DUF26 proteins in response to salt stress in *Salicornia* and other species. The interaction of SbiSALTY with ribosomes is another hypothesis that needs to be explored in more detail. RGG proteins are the second largest RNA binding family in humans and have been reported to interact with ribosomal proteins. Three RGG proteins in Arabidopsis were recently found to interact with proteins of the ribosomal subunits 40s and 60s. Accordingly, SbiSALTY has a high similarity in its coiled-coiled region to the 40s ribosomal protein S7. SbiSALTY possesses a high percentage of identity with the Arabidopsis RGG proteins, ~60%. Together with the observed ER localization of SbiSALTY in rice protoplasts, suggest that SbiSALTY might interact with ribosomes; however, further studies are required to test this hypothesis. We have included a more thorough discussion of SbiSALTY in the discussion (Page 35 lines 21-23, page 36 lines 12-15 and 21-23, and page 37 lines 1-2 and 6-11).

6. To identify genes of *S. bigelovii* involved in salt tolerance, genes were selected for salt tolerance assays just in yeast, the authors should examine the salt tolerance functionality of several genes in genetically transgenic materials with overexpression or CRISPR in *S. bigelovii* or Arabidopsis thaliana or rice.

We fully agree that the next step is to study the effects of these genes and test them in other plant systems. We currently focused on the development of *Salicornia* as a species to study salt tolerance and on the identification of possible mechanisms of salt tolerance. Molecular studies in *Salicornia* are very limited and we first needed to create the bases for *Salicornia* research. Due to the nature of studying an organism almost from scratch, we decided to test candidate genes in yeast, as we had more than 50 candidate genes. We are very interested in testing several hypotheses derived from this study and we will hopefully show more results in further studies. Unfortunately, a *Salicornia* transformation system has not been developed, making it currently impossible to do CRISPR knockouts in *S. bigelovii*. However, we do aim to test our results in future studies testing some of the candidate genes in other plant systems such as *Arabidopsis* or even crop species such as rice.

7. The authors need to verify that the figure legends provide all necessary details for the reader to understand the figures. A few examples- the ER marker in Figure 6g should detail which ER genes with which reference? Subcellular localization in yeast of Supplementary Figure 15/16/17/18 needs details and the picture arrangement should display in the same format (some 3 pictures and some 4)? Figure ordinate in Supplementary Figure 15c should align?

We have included the details of the ER marker and its reference in the figure legend of Figure 7 (previously Figure 6) and in methods. We appreciate bringing the discrepancies across yeast Supplementary Figures (now 20-22). We have normalized their format and now each gene displays 3 images in their second panel. The y-axis in Supplementary Figure 15c (now Supplementary Figure 20c) is aligned with the guidelines while the x-axis was tilted to fit the image, but all labels are equidistant to each other.

8. Citation of references, some titles may only have the first letter of the word capitalized, such as ref 6, 7, 9, while others may all have the first letter capitalized, such as ref13, 18, 20; some journal is abbreviated as ref7, 14, while others are spelled as ref12, 13.

We have adjusted the references format to display only the first letter as capitalized.

Reviewer #3

The manuscript examines *Salicornia* to understand its underlying ability to withstand high NaCl concentrations mainly by characterizing its transcriptome, followed by examining the effect of several proteins involved in salt response in a yeast model. Overall there is a considerable amount of work presented that is worthy of publication but I would not consider the work significant in that it feels preliminary with unfortunately only some insight into the molecular and cellular mechanisms.

We appreciate the comments from the reviewer. This work aimed to generate a solid baseline for *Salicornia* research, as very few molecular studies have been done for this genus. Due to a lack of studies in *Salicornia*, we had to build the molecular resources and carefully generate hypotheses through different approaches from scratch. We wholeheartedly agree that further experiments are required to fully understand the molecular mechanisms behind the extreme salt tolerance of *Salicornia*. We think that this study has identified a number of candidate genes and potential salt tolerance mechanisms in *Salicornia* that, with the help of future studies, might be important towards the understanding of salt tolerance in plants. To strengthen our hypotheses, we have now included in the manuscript a subcellular localization assay for SbiSOS1. This is described in more detail in our reply to the first comment of Reviewer 1.

One of the issues is the switching between several membrane associated proteins and one of the characterized soluble proteins by NMR. I understand that it is difficult to work with membrane proteins by NMR, but the emphasis in the early part of the results on membrane proteins and ion channels but then later switching to a soluble protein where the mechanism is unclear confused me. Also, the inclusion of NMR data and the observation that the protein is disordered did not show any significant mechanistic insight. The authors mention late embryogenesis proteins in the discussion, which are proteins known to be involved in stress tolerance, but do not appear to connect that to any of the identified proteins.

We understand that the shift from studying membrane proteins to the characterization of a soluble protein might be confusing. The aim of the study was to explore possible mechanisms of salt tolerance of *Salicornia bigelovii*, of which we identified (or hypothesized) at least 2. We knew that *Salicornia* accumulated high concentrations of Na⁺ in its shoots, hence, we hypothesized that transporters might be crucial to avoid cell toxicity. For this reason, we started the study focusing on membrane proteins, which led to our observation that SOS1 may be localized to the tonoplast. During our transcriptomics and proteomics analyses, we found another protein that piqued our interest, SbiSALTY. While this wasn't a membrane protein, it had an interesting profile and was strongly induced by salinity, so we decided to include it in our yeast experiments. To our surprise, it was highly beneficial for salt tolerance in yeast. For this reason, we decided to further characterize this protein. Very little information was available for this protein, so we decided to use NMR to confirm the intrinsically disordered nature of this protein. Nevertheless, this protein possesses a structured region, which is predicted to be able to form coiled-coils and is similar to the 40s ribosomal protein, S7. This potentially allows SbiSALTY to interact with ribosomes. We mentioned the late embryogenesis proteins as means to exemplify that some intrinsically disordered proteins have been linked to stress. We have now further expanded this discussion and made the link between intrinsically disordered proteins and RGG proteins. We also now elaborate on how RGG proteins have been linked to interactions with ribosomal proteins and hypothesized about their mode of action. We hope that the extra discussion increases the readability of the manuscript (Page 35 lines 21-23, page 36 lines 12-15 and 21-23, and page 37 lines 1-2 and 6-11).

The second issue is that the comparison of the genes that increased salt tolerance versus those that decreased it. Again, a considerable amount of work has been performed but I have a difficult time connecting it to a mechanism. In Figure 5, there are genes that increased tolerance and those that decreased it. My interpretation is that the expression of some of these genes may be up-regulated and others are down-regulated, but I did not see an explanation of this in the results. The authors also point out in the discussion that the lack of protection in yeast may be due to different localization compared to plants. These results could be strongly supported by several selective knockout studies in *Salicornia* to show that these proteins are actually important for salt tolerance.

We think that there could be multiple reasons for why some genes increase or decrease salt tolerance in yeast. The first one is, as the reviewer suggested, that some genes might be upregulated while others are downregulated. However, some genes, such as the *NHXs*, did not change in expression across treatments, and different *NHXs* increased or decreased salt tolerance in yeast. Another example is *SbiENT1*, which was upregulated in *S. bigelovii* upon increased salinity but decreased yeast salt tolerance. We think that another reason for the differences in yeast phenotype is the fact that yeast is a unicellular organism. Plants have different tissues and the accumulation of Na^+ and Cl^- in certain cell-types could prevent it from accumulating in the photosynthetically active cells. Moreover, different tissues have different functions, and some genes might be beneficial in some while detrimental in others. Furthermore, as yeast is a unicellular organism, any additional accumulation of Na^+ would be detrimental. We also hypothesize that the subcellular localization of proteins in yeast might sometimes differ in plants and the negative effects of mis-localization can't be discarded. We have included this into the discussion and will hopefully make it more clear for the readers (Page 35 lines 11-17). Unfortunately, there is no transformation technique developed in *Salicornia*, making it impossible to generate selective knockout studies. To overcome this, future studies generating random mutations, for example with EMS, could help identify key genes involved in the salt tolerance of *S. bigelovii*.

Reviewer #1 (Remarks to the Author):

The authors have made important progress in this revision and have addressed many of my concerns. I appreciate the addition of tobacco transient assays to determine the localization of the SbiSOS protein to test whether it shows the anticipated tonoplast localization that their proteomics analysis suggests. Unfortunately, the characterization of SbiSOS localization was not described well and is incomplete. It seems from the methods section that plasmolysis was performed, but the rationale for the treatment was not discussed, nor were images shown for non-plasmolysed samples. Most importantly, the localization of SbiSOS should be compared to AtSOS1, to determine if the SbiSOS protein shows differential localization. The methods section alluded to such an experiment, but I don't see these results presented in the manuscript or Fig 5. Since this is one of the more significant findings of the manuscript, it is important that the authors perform these experiments as thoroughly as possible. Currently the data do not strongly support a different localization pattern for SbiSOS compared to AtSOS1.

1) The revised title should remove the term "remarkable". To indicate that the species exhibits high levels of salinity tolerance, the authors could use a more standard term such as halophyte or extremophyte.

2) The data presented in Figure 5 is not sufficient to determine the subcellular localization of the SbiSOS1 protein. It is not described what is meant by "membrane subcellular localization assay". Some samples appear to be plasmolyzed with hechtian strands labeled, but this is not clear if the samples have been osmotically shocked or not, as described. Plasma membrane localization is normally confirmed by plasmolysis, to separate the plasma membrane from the cell wall, but I assume that would have been done in experiments where SbiSOS-eGFP is co-localized with mCherry-AtPIP1;4, but that doesn't appear to be the case.

3) Page 50, line 18: The authors state in the methods section that AtSOS1 localization was also examined together with SbiSOS, however, I don't see where that data is. Indeed, the authors should contrast the localization of both AtSOS1 and SbiSOS to show that the localization in tobacco is able to recapitulate the expected difference in subcellular localization.

4) The abstract describes the localization of SbiSOS at the tonoplast, but the data presented suggests a dual localization. I agree that the plasma membrane localization could be an artifact of overexpression, but without confirmation by some other method, the authors should list both cellular compartments as potential sites for SbiSOS localization.

5) I would advise the authors to change the main figures to be color-blind compatible rather than leaving this in the supplementary figures. Similarly, figure 7 should be adjusted as well.

6) Page 22, line 1: Please revise sentence "suggests for a mechanism in Salicornia to reduce cytosolic salt accumulation" for clarity.

Reviewer #2 (Remarks to the Author):

The revised version of the manuscript has much improved and my previous concerns have largely been addressed.

Reviewer #3 (Remarks to the Author):

The authors have addressed my concerns to a reasonable level.

Response to the Reviews

We would like to thank the reviewers again for their continuous time and effort put into reviewing our manuscript. We appreciate their suggestions, as they improve the quality of the manuscript.

Reviewer #1 (Remarks to the Author):

The authors have made important progress in this revision and have addressed many of my concerns. I appreciate the addition of tobacco transient assays to determine the localization of the SbiSOS protein to test whether it shows the anticipated tonoplast localization that their proteomics analysis suggests. Unfortunately, the characterization of SbiSOS localization was not described well and is incomplete. It seems from the methods section that plasmolysis was performed, but the rationale for the treatment was not discussed, nor where images shown for non-plasmolysed samples. Most importantly, the localization of SbiSOS should be compared to AtSOS1, to determine if the SbiSOS protein shows differential localization. The methods section alluded to such an experiment, but I don't see these results presented in the manuscript or Fig 5. Since this is one of the more significant findings of the manuscript, it is important that the authors perform these experiments as thoroughly as possible. Currently the data do not strongly support a different localization pattern for SbiSOS compared to AtSOS1.

We thank the reviewer for the suggestions and we have tried to address them to the best of our ability. We address the specific comments below:

1) The revised title should remove the term "remarkable". To indicate that the species exhibits high levels of salinity tolerance, the authors could use a more standard term such as halophyte or extremophyte.

We have removed the term remarkable and included the term extremophyte in the title. It now reads: "Insights into the salinity tolerance of the extremophyte *Salicornia bigelovii*: roles for the RGG protein SALTY, and neo-localization of SOS1 to the tonoplast"

2) The data presented in Figure 5 is not sufficient to determine the subcellular localization of the SbiSOS1 protein. It is not described what is meant by "membrane subcellular localization assay". Some samples appear to be plasmolyzed with hechtian strands labeled, but this is not clear if the samples have been osmotically shocked or not, as described. Plasma membrane localization is normally confirmed by plasmolysis, to separate the plasma membrane from the cell wall, but I assume that would have been done in experiments where SbiSOS-eGFP is co-localized with mCherry-AtPIP1;4, but that doesn't appear to be the case.

We have adjusted the figure legend of Figure 5 to improve its readability and we have indicated that the samples were treated with 300 mM mannitol to induce plasmolysis (previously only found in methods). We have now highlighted hechtian strands in all panels to increase visibility of the plasma membrane. The colocalization of SbiSOS1-eGFP with mCherry-AtPIP1;4 can be

found in Figure 5d-f. We have also now included, as suggested, panels showing AtSOS1 localization to serve as comparison to SbiSOS1 localization.

3) Page 50, line 18: The authors state in the methods section that AtSOS1 localization was also examined together with SbiSOS, however, I don't see where that data is. Indeed, the authors should contrast the localization of both AtSOS1 and SbiSOS to show that the localization in tobacco is able to recapitulate the expected difference in subcellular localization.

We have included in Figure 5 the localization assays of AtSOS1 in tobacco leaves to contrast the localization of SbiSOS1 and AtSOS1. We hope that this inclusion helps to provide a better overview of SbiSOS1.

4) The abstract describes the localization of SbiSOS at the tonoplast, but the data presented suggests a dual localization. I agree that the plasma membrane localization could be an artifact of overexpression, but without confirmation by some other method, the authors should list both cellular compartments as potential sites for SbiSOS localization.

We agree that the possibility of the dual localization of SbiSOS1 to the plasma membrane and the tonoplast should be discussed. We have added the word "include" in the abstract and the possible dual localization is discussed in page 21 lines 16-23, page 22 lines 1-6, page 34 lines 19-23 and page 35 lines 1-2.

5) I would advise the authors to change the main figures to be color-blind compatible rather than leaving this in the supplementary figures. Similarly, figure 7 should be adjusted as well.

We completely agree and we had been trying to find a work-around. The "green-red-images" ensure best contrast between different markers, as it is hard to find a good balance in composite images, particularly in overlapping regions. We think this is why in most main figures in Nature Communications we can find this colour layout. For this reason, we think that it might be better to have two different sets of images. We have included color-blind compatible Figure versions in our Supplementary Figures (Figure 5 -> Supplementary Figure 12, Figure 7 -> Supplementary Figure 23, and Supplementary Figure 13 -> Supplementary Figure 14).

6) Page 22, line 1: Please revise sentence "suggests for a mechanism in Salicornia to reduce cytosolic salt accumulation" for clarity.

We have modified the sentence and we hope it is clearer. It can be found in page 22 lines 1-3.

Reviewer #2 (Remarks to the Author):

The revised version of the manuscript has much improved and my previous concerns have largely been addressed.

We thank the reviewer for the work put in the revision of this manuscript.

Reviewer #3 (Remarks to the Author):

The authors have addressed addressed my concerns to a reasonable level.

We thank the reviewer for the work put in the revision of this manuscript.

Reviewer #1 (Remarks to the Author):

The authors have now sufficiently addressed my concerns. I congratulate the authors on a putting together an illuminating manuscript that opens up the exploration of new mechanisms for stress tolerance in plants.